# Nuclear receptor NR4A is required for patterning at the ends of the planarian anterior-posterior axis

Dayan J Li[1,2,3,4], Conor L McMann[1,2,3], Peter W Reddien[1,2,3]*

[1]Whitehead Institute for Biomedical Research, Cambridge, United States; [2]Department of Biology, Massachusetts Institute of Technology, Cambridge, United States; [3]Howard Hughes Medical Institute, Chevy Chase, United States; [4]Harvard/MIT MD-PhD Program, Harvard Medical School, Boston, United States

**Abstract** Positional information is fundamental to animal regeneration and tissue turnover. In planarians, muscle cells express signaling molecules to promote positional identity. At the ends of the anterior-posterior (AP) axis, positional identity is determined by anterior and posterior poles, which are putative organizers. We identified a gene, *nr4A*, that is required for anterior- and posterior-pole localization to axis extremes. *nr4A* encodes a nuclear receptor expressed predominantly in planarian muscle, including strongly at AP-axis ends and the poles. *nr4A* RNAi causes patterning gene expression domains to retract from head and tail tips, and ectopic anterior and posterior anatomy (e.g., eyes) to iteratively appear more internally. Our study reveals a novel patterning phenotype, in which pattern-organizing cells (poles) shift from their normal locations (axis extremes), triggering abnormal tissue pattern that fails to reach equilibrium. We propose that *nr4A* promotes pattern at planarian AP axis ends through restriction of patterning gene expression domains.

DOI: https://doi.org/10.7554/eLife.42015.001

*For correspondence:
reddien@wi.mit.edu

**Competing interests:** The authors declare that no competing interests exist.

## Introduction

Metazoans display a large diversity of developmental modes and adult forms. Processes that govern the generation of form, collectively known as patterning, act to regulate cell identity, location, and number (*Wolpert, 1969*). The mechanisms by which pattern is established and maintained in adult tissues, however, are poorly understood. Planarians are freshwater flatworms capable of remarkable feats of whole-body regeneration and offer the opportunity, as a model system, to generate important insights into the molecular and cellular mechanisms that can generate and maintain adult tissue pattern.

The planarian anterior-posterior (AP) axis has been a target for study of positional information in adult biology. RNA interference (RNAi) approaches have identified genes with regionalized expression domains that are important in AP patterning (*Forsthoefel and Newmark, 2009*; *Adell et al., 2010*; *Reddien, 2011*). Genes with such constitutive regionalized expression and association with pathways with planarian patterning roles are called position control genes or PCGs (*Witchley et al., 2013*; *Scimone et al., 2016*; *Fincher et al., 2018*). PCGs are predominantly expressed in planarian muscle (*Witchley et al., 2013*; *Scimone et al., 2016*; *Fincher et al., 2018*; *Scimone et al., 2018*). A number of PCGs encode evolutionarily conserved signaling proteins, such as members of the FGFRL family and Wnt/ß-catenin pathway components (*Reddien, 2011*). For example, inhibition of the *FGFRL* gene *ndl-3* and the *Wnt* gene *wntP-2* led to the formation of ectopic mouths and pharynges in the trunk (*Lander and Petersen, 2016*; *Scimone et al., 2016*). These findings support a model in which planarian muscle serves as a source of positional information that patterns surrounding tissues,

**eLife digest** Many animals are able to regenerate tissue that has been lost through illness or injury. Flatworms called planarians have long been used to study tissue regeneration because of their remarkable ability to completely regenerate their whole body from small pieces of tissue. Furthermore, the stem cells of adult planarians continually produce new cells to replace dying cells in a process called tissue turnover.

For regeneration and tissue turnover to be successful, it is important for the new cells to form in the right location in the body; for example, new eye cells need to form in the head. Genes known as position control genes are active in muscle at specific locations along the body of a flatworm to regulate both regeneration and tissue turnover. However, it was not clear how these genes coordinate with stem cells to produce new cells in the correct positions in the body.

Li et al. examined the effects of a gene known as *nr4A* that is particularly active in muscle at the head and tail ends of planarians. Using a technique called RNA interference to decrease the activity of *nr4A* in planarians disrupted the patterns of tissues at each end of the flatworms. Over time, the activity of the position control genes also became restricted to locations progressively farther away from the head and tail. As a result, cells that were intended to replace tissues in the head or tail were deposited increasingly far away from these locations. For example, new eyes formed repeatedly in the planarians, with each set farther away from the head tip than the last. Li et al. propose that these disruptions of normal tissue patterning ensue because the cells that organize such patterns at the ends of the planarian (the poles) are themselves misplaced within the existing body pattern.

The *nr4A* gene can be found in a wide range of animal species. Understanding how this gene affects tissue patterns in planarians could therefore also help researchers to discover how adult tissue patterns form and are maintained in animals more generally.

DOI: https://doi.org/10.7554/eLife.42015.002

for instance by influencing the specification of resident stem cells called neoblasts and/or the localization of the specified progeny cells of neoblasts (*Reddien, 2011*; *Witchley et al., 2013*; *Scimone et al., 2017*; *Wurtzel et al., 2017*; *Atabay et al., 2018*; *Hill and Petersen, 2018*).

Some PCGs are expressed in small groups of cells at the extreme ends of the primary axis called the anterior and posterior poles. Organizers, commonly studied in embryogenesis, are groups of cells that influence the fate of their neighboring cells for the production of tissue pattern (*Spemann and Mangold, 1924*; *Lemaire and Kodjabachian, 1996*; *De Robertis et al., 2000*). Planarian poles function as putative organizers in planarian head and tail patterning (*Scimone et al., 2014*; *Vogg et al., 2014*; *Owlarn and Bartscherer, 2016*; *Oderberg et al., 2017*; *Reddien, 2018*). The anterior pole expresses *notum*, which encodes a Wnt inhibitory protein (*Petersen and Reddien, 2011*), the transcription factor (TF)-encoding genes *foxD*, *zic-1*, *islet*, *pitx* (*Hayashi et al., 2011*; *Currie and Pearson, 2013*; *März et al., 2013*; *Scimone et al., 2014*; *Vásquez-Doorman and Petersen, 2014*; *Vogg et al., 2014*), and *follistatin*, which encodes an Activin-inhibitory protein (*Gaviño et al., 2013*; *Roberts-Galbraith and Newmark, 2013*). The posterior pole expresses *wnt1* (*Petersen and Reddien, 2008*), the TF-encoding genes *islet1* (*Hayashi et al., 2011*; *März et al., 2013*) and *pitx* (*Currie and Pearson, 2013*; *März et al., 2013*), and *wnt11-2* (*Adell et al., 2009*; *Gurley et al., 2010*). Ablation of the anterior pole through inhibition of pole-specifying genes resulted in head patterning defects, including cyclopia and fused brain lobes (*Felix and Aboobaker, 2010*; *Blassberg et al., 2013*; *Chen et al., 2013*; *Currie and Pearson, 2013*; *März et al., 2013*; *Scimone et al., 2014*; *Vásquez-Doorman and Petersen, 2014*; *Vogg et al., 2014*). Ablation of the posterior pole or inhibition of posterior-pole PCGs resulted in tail-patterning defects, such as stunted tails and fused ventral nerve cords (*Petersen and Reddien, 2009*; *Adell et al., 2009*; *Gurley et al., 2010*; *Hayashi et al., 2011*; *Currie and Pearson, 2013*; *März et al., 2013*).

We used tissue-fragment and single-cell RNA sequencing to determine the anterior- and posterior-pole transcriptomes, respectively, and expanded the known repertoire of genes with enriched expression in these putative organizers. Inhibition of identified genes uncovered a novel patterning role for the gene *nr4A*, which encodes a planarian ortholog of the broadly conserved NR4A nuclear

receptor. *nr4A* is expressed broadly in muscle, including strongly in both poles and other muscle cells at the ends of the AP axis. *nr4A* inhibition results in a novel patterning abnormality in which PCG expression domains shift away from the head and tail tips, followed by similar shifts in differentiated tissues during homeostatic turnover in uninjured animals. The *nr4A* phenotype represents a patterning abnormality in which a stable anatomical pattern is not reached and instead iterative duplication of tissues continues to occur. This phenotype reveals how homeostatic placement of pattern-generating cells at a correct location is essential for the maintenance of stable tissue pattern. We conclude NR4A is a transcription factor that helps regulate tissue pattern at both ends of the AP body axis.

## Results

### RNA sequencing identifies new genes expressed at the anterior and posterior poles

To comprehensively identify genes expressed in the planarian poles, we performed RNA sequencing of anterior and posterior poles from both uninjured animals and regenerating trunks 72 hr post amputation (hpa). To obtain the anterior-pole transcriptome, we surgically isolated anterior poles (*Figure 1A*). Differential gene expression analysis identified 203 genes with significantly higher ($p_{adj}$ <0.05) expression in pole-containing pieces compared to flanking pieces from uninjured animals (*Figure 1A*, *Supplementary file 1A*). In the anterior blastema (an unpigmented outgrowth of differentiating tissue), 86 genes had pole-enriched expression ($p_{adj}$ <0.05) (*Figure 1A*, *Supplementary file 1B*). 51 genes had enriched expression in the poles of both uninjured and regenerating animals and these included the previously published anterior-pole genes *foxD*, *notum*, *zic-1*, and *follistatin* (*Figure 1A*, *Supplementary file 1A, B*).

Because the posterior pole is diffuse, its excision contains substantial non-pole tissue. Therefore, we utilized single-cell RNA sequencing of cells from uninjured tail tips and from 72 hpa posterior blastemas to obtain the posterior-pole transcriptome (*Figure 1—figure supplement 1A*). We used qRT-PCR to identify FACS-isolated single cells expressing *wnt1* and the muscle marker *collagen* to identify pole cells (*wnt1*$^+$; *collagen*$^+$) and non-pole muscle cells (*wnt1*$^-$; *collagen*$^+$) (*Figure 1—figure supplement 1A*). From 768 cells, 11 posterior pole cells were identified (six from uninjured animals and five from blastemas); 90 non-pole muscle cells (43 from uninjured animals and 47 from blastemas) were used as controls (*Figure 1—figure supplement 1B*). Using single-cell differential expression (SCDE) analysis (*Kharchenko et al., 2014*), we identified 198 genes with significantly higher expression (p<0.05) in posterior-pole cells compared to non-pole posterior muscle cells (*Figure 1—figure supplement 1B*, *Supplementary file 1C*). *wnt1*, *pitx*, and *islet1* were in the top 16% of those 198 genes ranked by enrichment in posterior-pole cells (*Supplementary file 1C*).

We assessed the in vivo expression of 133 candidate anterior-pole genes and 96 candidate posterior pole genes by whole-mount in situ hybridization (WISH) and identified 12 new genes expressed in the anterior pole (*Figure 1B*) and 10 new genes expressed at the posterior pole (*Figure 1—figure supplement 1C*). The new pole genes included those encoding a predicted secreted factor (*kallmann1*), cell-surface receptors (*ror1*, *ephr4*, *ephr5*, *pcdh9*, *dcc*, *ddr2*), and transcription factors (*islet2*, *musculin*, *nr4A*) (*Figure 1B*, *Figure 1—figure supplement 1C*). Interestingly, some of these genes, such as *kallmann1*, *ephr5*, and dd_20026 were expressed in both anterior and posterior poles, identifying molecular similarity between these structures (*Figure 1B*, *Figure 1—figure supplement 1C*, *Supplementary file 1A-C*). We confirmed the expression of the genes in anterior pole cells by their co-expression with the anterior pole markers *notum* and *foxD* (*Figure 1C*).

### *nr4A* is required for head and tail patterning

We utilized RNAi to determine the functions of new pole genes and identified a novel phenotype following inhibition of the gene *nr4A*. *nr4A* encodes the *S. mediterranea* homolog to proteins of the NR4A nuclear receptor family, which are found broadly in metazoans (*Bridgham et al., 2010*; *Escriva et al., 2004*) (*Figure 2—figure supplement 1*). RNAi of *nr4A* resulted in the progressive formation of posterior ectopic eyes in uninjured animals (*Figure 2A*). After initiation of *nr4A* RNAi, the distance between the original eyes and the head tip progressively decreased. By 6 weeks of *nr4A* RNAi, an additional eye pair emerged just posterior to the original eyes. As time progressed, the

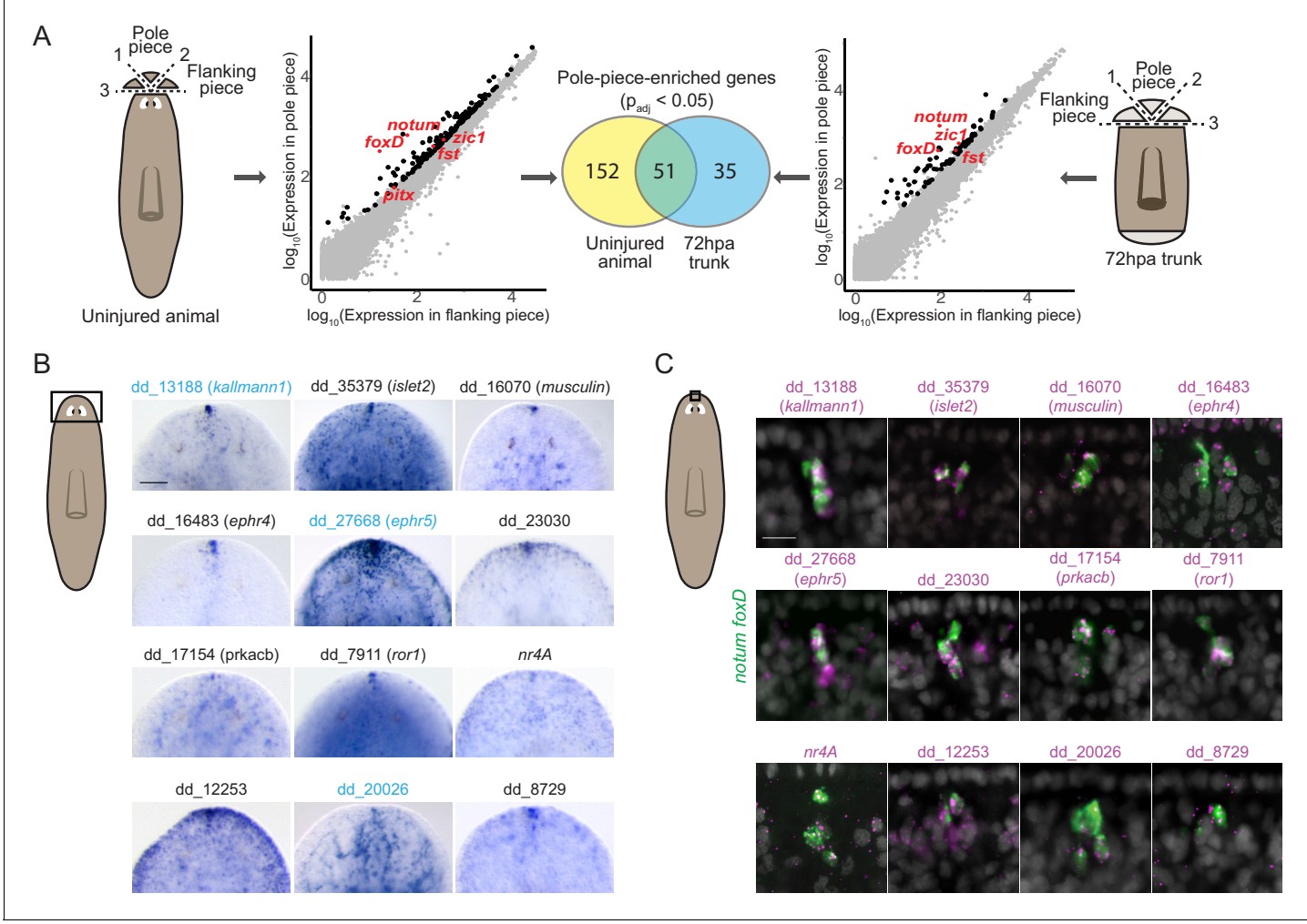

**Figure 1.** RNA sequencing of head-tip fragments identifies new anterior-pole-enriched genes. (**A**) Anterior pole bulk RNA sequencing: RNA sequencing of excised fragments containing the anterior pole ('Pole piece') and adjacent fragments without the pole ('Flanking piece') from uninjured animals and anterior blastemas of regenerating trunks 72 hr post amputation (72hpa) was performed (cartoons, dotted lines are cut planes, and dotted line numbers indicate cut sequence). Scatter plots show results from differential expression analysis, which identified genes with enriched expression ($p_{adj}$ <0.05) in pole pieces compared to flanking pieces (black data points), including previously published anterior pole genes (red data points). Many genes with enriched expression were identified in both uninjured and 72hpa animals, as shown in the Venn diagram. (**B**) Head expression of anterior pole candidates from RNA sequencing by whole-mount in situ hybridization. Gene names, if present, represent best human BLAST hits; numerical names indicate the Smedv4.1 Dresden transcriptome assembly transcript number (See 'Gene nomenclature' in Materials and methods). Genes in blue were also expressed in the posterior-pole region (*Figure 1—figure supplement 1*). Area imaged is indicated by the box in the cartoon on the left. Scale bar represents 100 µm. (**C**) Co-expression of anterior pole gene candidates with pooled pole markers *notum* and *foxD* by fluorescence in situ hybridization. DAPI nuclear stain in gray. Area imaged is indicated by the box in the cartoon on the left. Scale bar represents 10 µm. Images are representative of results seen in at least four animals.

DOI: https://doi.org/10.7554/eLife.42015.003

The following figure supplement is available for figure 1:

**Figure supplement 1.** Single-cell RNA sequencing of tail tip fragments identifies new genes expressed in the posterior pole region.

DOI: https://doi.org/10.7554/eLife.42015.004

original and second eye pairs became closer to the head tip and decreased in size. A third eye pair, posterior to the second eye pair then appeared, resulting in animals with six eyes. This iterative process continued with ongoing *nr4A* inhibition, generating animals with as many as 10 eyes after 16 weeks of RNAi.

By 12 weeks of *nr4A* RNAi, uninjured animals developed dorsal outgrowths at their tail tips, suggesting that *nr4A* influences both head and tail tip pattern (*Figure 2B*). After transverse amputation

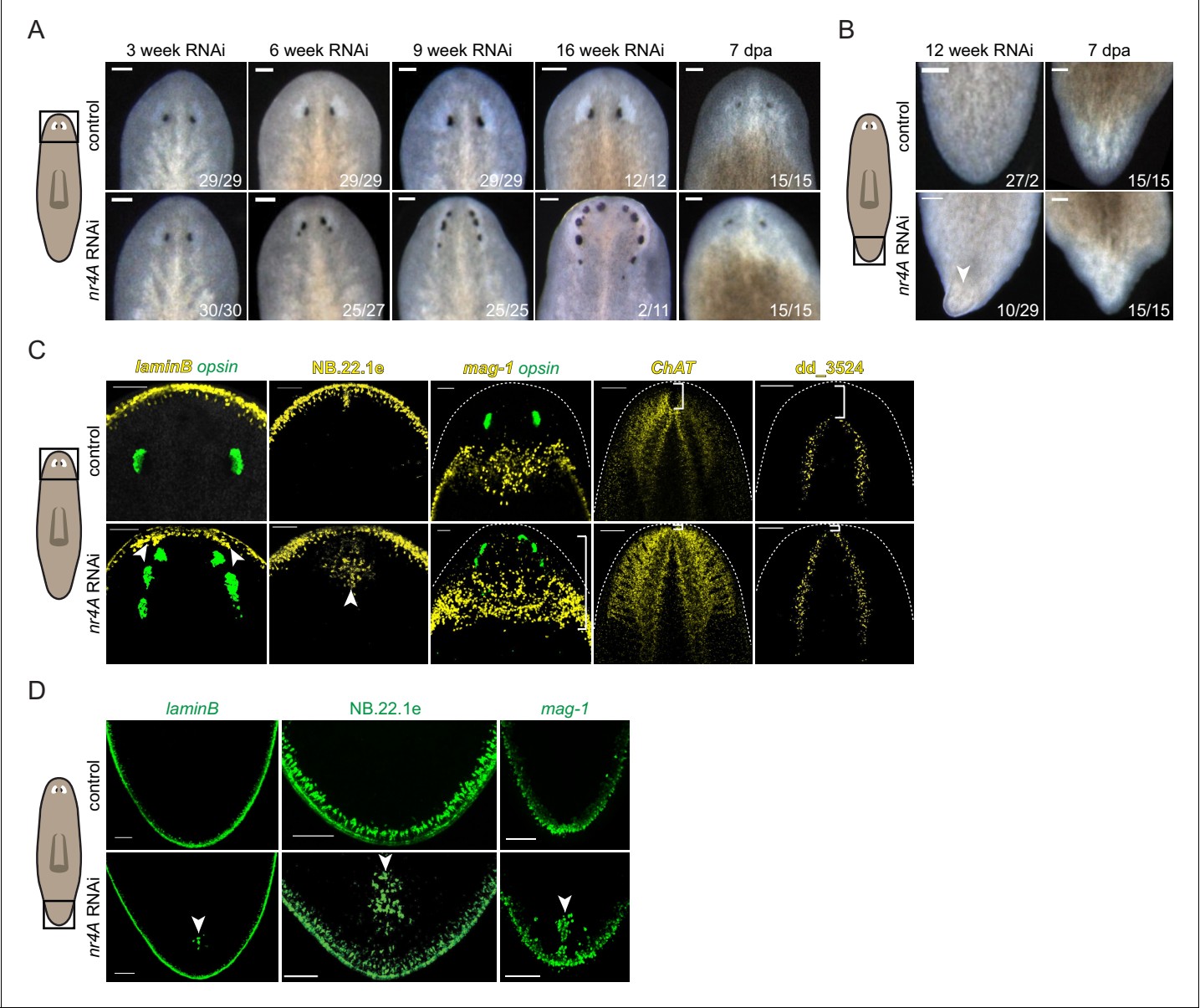

**Figure 2.** *nr4A* RNAi leads to head and tail patterning defects. (**A**) Images of the heads of live animals at different time points during control and *nr4A* RNAi. Progressive development of posterior ectopic eyes and complete head regeneration at 7 days post amputation (seven dpa) were observed in *nr4A* RNAi. Proportions indicate numbers of animals with phenotype shown over total number of animals. Scale bars represent 150 μm. (**B**) Images of the tails of live animals at 12 weeks of RNAi. Development of an outgrowth at the tail tip and complete tail regeneration without an outgrowth at seven dpa were observed in *nr4A* RNAi. Scale bars represent 150 μm. (**C**) Differentiated tissue marker expression in the head of 9 week RNAi animals by FISH. Ectopic differentiated tissue marker expression is indicated by arrow heads; posteriorly expanded *mag-1*+ cells and head tip loss are indicated with brackets. Dotted white lines indicate anterior borders of the heads. Scale bars for images with *laminB*, NB.22.1e, and *mag-1* represent 100 μm; scale bar for images with *ChAT* and dd_3524 represent 200 μm. (**D**) FISH with differentiated tissue markers showed ectopic marker expression in the tails of 9 week *nr4A(RNAi)* animals (arrow heads) compared to 9 week control animals. Scale bars represent 100 μm. Areas imaged are indicated by the box in the cartoons on the left for all panels. Fluorescence images are maximum intensity projections. All images are representative of results seen in at least four animals.

DOI: https://doi.org/10.7554/eLife.42015.005

The following figure supplements are available for figure 2:

**Figure supplement 1.** Phylogenetic analysis of *Schmidtea mediterranea* NR4A.

DOI: https://doi.org/10.7554/eLife.42015.006

**Figure supplement 2.** *nr4A* RNAi leads to the appearance of ectopic *laminB*+ cells posterior to the dorsal-ventral (DV) boundary of the head tip.

DOI: https://doi.org/10.7554/eLife.42015.007

of heads and tails, *nr4A(RNAi)* trunk fragments were capable of regenerating heads with normal eye number and tails without outgrowths (*Figure 2A,B*), indicating that the *nr4A* phenotype is most readily apparent during tissue turnover associated with maintenance of anatomical pattern.

Using tissue-specific RNA probes and fluorescence in situ hybridization (FISH), we detected patterning abnormalities of multiple differentiated tissues in both the head and the tail of 9 week *nr4A (RNAi)* animals. In addition to ectopic eyes, labeled by an RNA probe to *opsin*, the heads of *nr4A (RNAi)* animals developed internal (along the AP axis) ectopic foci of epidermal cells normally restricted to the animal periphery at the dorsal-ventral (DV)-median plane (*laminB*[+] and NB.22.1e[+] cells) (*Figure 2C*, *Figure 2—figure supplement 2*). Marginal adhesive gland cells (*mag-1*[+]) are normally present in a prepharyngeal domain posterior to the eyes and along the body margin posteriorly from this zone (*Zayas et al., 2010*) (*Figure 2C*). The distance from the original eyes to the posterior end of the *mag-1*[+] domain was significantly greater in *nr4A(RNAi)* animals than in control animals (*Figure 2C*, *Supplementary file 1D*). FISH with neural markers *ChAT* and dd_3524, which are expressed in the inverted 'U'-shaped cephalic ganglia, showed a loss of the head region between the apex of the head and the anterior aspect of the brain following *nr4A* RNAi (*Figure 2C*). This phenotype is consistent with the shortening of the distance between the original eyes and the apex of the head, and the iterative appearance of ectopic posterior eyes with time. These results suggest that *nr4A* inhibition causes a loss of head tip pattern involving shortening of the distance between the anterior end of differentiated tissues and the head tip and posterior expansion of anteriorly restricted tissues.

Similar to the changes we observed in the head, ectopic foci of DV boundary epidermal cells (*laminB*[+] and NB.22.1e[+]) appeared internally (along the AP axis) in the tails of *nr4A(RNAi)* animals (*Figure 2D*). The secretory *mag-1*[+] cells normally restricted to the body margin around the tail periphery also appeared internally along the midline of tails following *nr4A* RNAi (*Figure 2D*), resulting in aberrant tail-tip patterning. These internally mis-localized tissues indicate a similarity between the defects at the anterior and posterior extremities. The mislocalization of differentiated tissues in *nr4A(RNAi)* animals demonstrates that *nr4A* activity is required for normal restriction of tissues to their proper domains at both extreme ends of the AP axis.

## *nr4A* inhibition causes shifts in multiple anterior-posterior PCG expression domains

The patterning defects in *nr4A(RNAi)* animals led us to examine the expression of patterning genes (PCGs) in these animals. The expression of several genes like *ndl-4*, *sFRP-1*, *notum*, and *foxD* at the anterior end of the AP axis defines the head extremity (*Figure 3A*). In uninjured *nr4A(RNAi)* animals analyzed at 9 weeks of RNAi, these expression domains were retracted from the head tip and expanded posteriorly (*Figure 3A*, *Figure 3—figure supplement 1*, *Supplementary file 1D*). *foxD*[+]; *notum*[+] anterior pole cells were no longer present in a cluster at the head tip, but instead were found scattered between the eyes (*Figure 3A*).

Similar to the anterior pole domain, the expression domains of PCGs that are broadly expressed in the head and pre-pharyngeal regions, *ndl-2*, *ndl-5*, *ndl-3*, and *wnt2*, were also shifted. Specifically, the anterior boundary of their expression domains was shifted posteriorly from the head tip following *nr4A* RNAi (*Figure 3A*, *Figure 3—figure supplement 1*, *Supplementary file 1D*). When considered relative to an anatomical landmark - the original eyes - the anterior boundaries of *ndl-2* and *ndl-5* expression domains, for example, were significantly shifted more posteriorly compared to those in control animals (*Figure 3A*, *Figure 3—figure supplement 1*, *Supplementary file 1D*). *ndk* expression was not excluded from the head tip in *nr4A(RNAi)* animals, but was also expanded more posteriorly (*Figure 3A*, *Figure 3—figure supplement 1*, *Supplementary file 1D*). Whereas *nr4A* RNAi impacted the expression of many head PCGs, inhibition of head PCGs *fz5/8-4*, *ndk*, *wntA*, and *foxD*, which are required for anterior patterning (*Cebrià et al., 2002*; *Scimone et al., 2014*; *Vogg et al., 2014*; *Scimone et al., 2016*), did not lead to detectable *nr4A* expression changes (*Figure 3—figure supplement 2*). Amputated *nr4A(RNAi)* animals were able to regenerate anterior poles by 72hpa (*Figure 3B*). However, at 7 days post amputation (7dpa) the anterior-pole-cell cluster was absent from the head tip and instead was disorganized and shifted posteriorly (*Figure 3B*). This anterior-pole positioning phenotype is similar to the anterior-pole phenotype we observed in uninjured *nr4A* RNAi (*Figure 3A*).

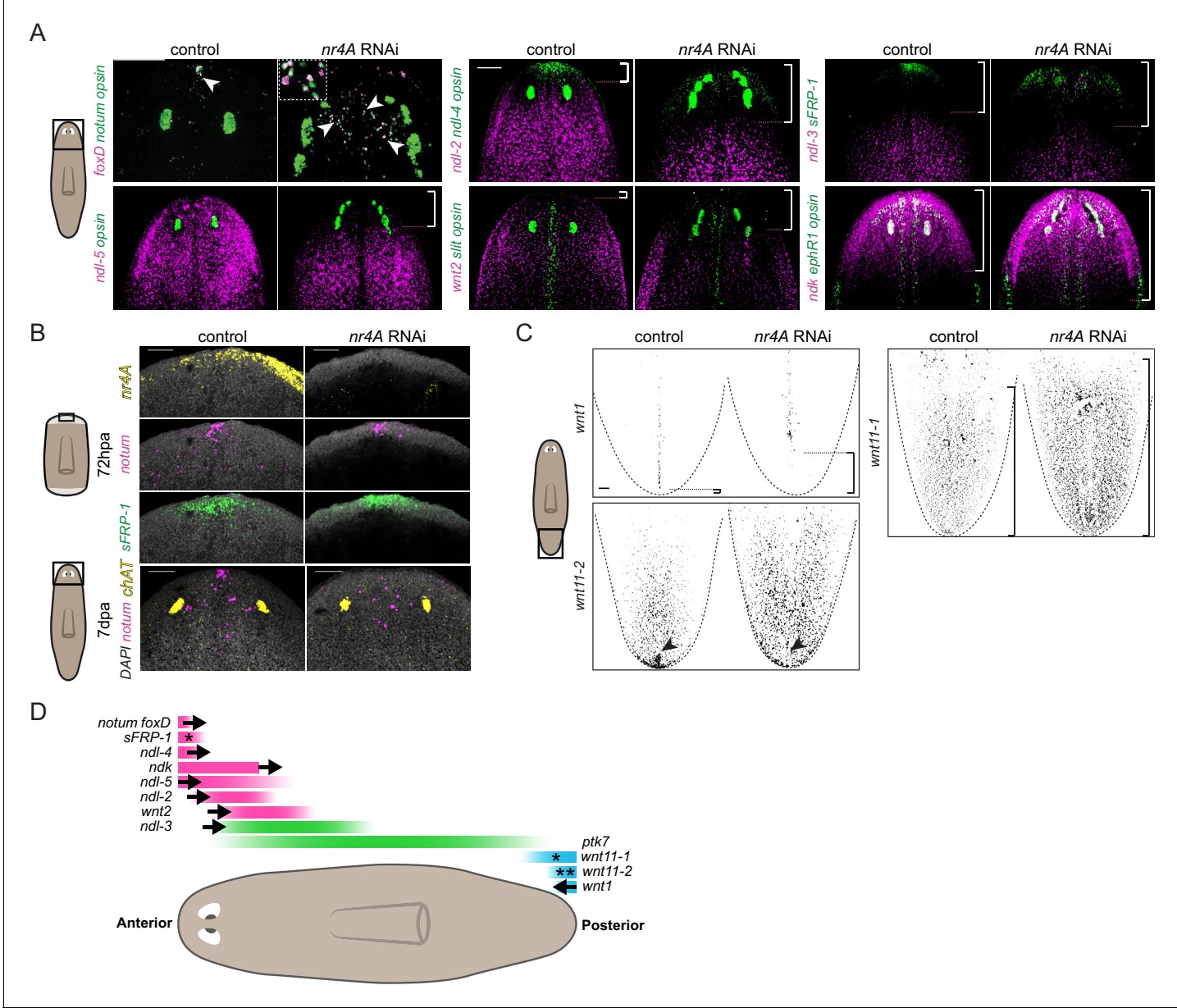

**Figure 3.** *nr4A* maintains PCG expression at the head and tail tips. (**A**) Eye marker *opsin* and head PCG expression by FISH in control and *nr4A(RNAi)* animals at 9 weeks of RNAi. Inset shows a magnified view of ectopic *notum*+; *foxD*+ anterior pole cells in *nr4A(RNAi)* animal. Scale bars represent 200 µm. (**B**) Anterior pole regeneration at the anterior blastema at 72 hr post amputation (72hpa) and 7 days post amputation (7dpa) by FISH. DAPI nuclear stain in gray. Scale bars represent 100 µm. (**C**) Tail PCG expression by FISH in control and *nr4A(RNAi)* animals at 9 weeks of RNAi. Scale bar represents 100 µm. (**D**) Diagram summarizing PCG domain shifts in *nr4A* RNAi. Statistically significant shifts of anterior or posterior PCG boundaries in the direction of the arrows are shown. A single asterisk denotes statistically significant area change without significant domain boundary shift. Double asterisks in *wnt11-2* domain denote the disappearance of its posterior-most expression cluster in *nr4A* RNAi. Quantification of PCG expression domain changes is included in *Figure 3—figure supplement 1* and *Supplementary file 1D*. Area imaged is indicated by the box in the cartoon on the left for all panels. Images are maximum intensity projections. All images are representative of results seen in at least four animals.

DOI: https://doi.org/10.7554/eLife.42015.008

The following figure supplements are available for figure 3:

**Figure supplement 1.** Quantification of shifts in PCG and *mag-1* expression domains in *nr4A* RNAi.
DOI: https://doi.org/10.7554/eLife.42015.009

**Figure supplement 2.** *nr4A* expression is independent of anterior PCGs.
DOI: https://doi.org/10.7554/eLife.42015.010

**Figure supplement 3.** Mid-body, DV, and midline pattern are largely intact in *nr4A* RNAi.

*Figure 3 continued on next page*

*Figure 3 continued*

DOI: https://doi.org/10.7554/eLife.42015.011

In the posterior, *wnt1* is normally expressed linearly along the midline of the tail tip (the posterior pole). At 9 weeks post-initiation of *nr4A* RNAi, the posterior-most expression domain of *wnt1* was absent, leaving a gap between the tip of the tail and *wnt1* expression (*Figure 3C*, *Figure 3—figure supplement 1*, *Supplementary file 1D*). Furthermore, the localized expression of *wnt11-2* at the tail tip was also abrogated by *nr4A* RNAi (*Figure 3C*, *Supplementary file 1D*). We also observed an increase in the area of *wnt11-1* expression in the tails of *nr4A(RNAi)* animals (*Figure 3C*, *Figure 3—figure supplement 1*, *Supplementary file 1D*).

In the midbody region, *nr4A* RNAi did not significantly affect PCG expression domains. Specifically, the location of the posterior boundaries of the expression domains of *ndl-2*, *ndl-3*, *ndl-5*, and *wnt2* were normal compared to the control (*Figure 3—figure supplement 3A*, *Supplementary file 1D*), despite the anterior boundaries of their expression domains being shifted (*Figure 3A*, *Figure 3—figure supplement 1*, *Supplementary file 1D*). Additionally, the length of *ptk7* expression domain, defined by the distance between its anterior and posterior boundaries in the midbody, was not significantly altered by *nr4A* RNAi (*Figure 3—figure supplement 3A*, *Supplementary file 1D*).

We also examined PCG expression pattern on the DV and medial-lateral axes of *nr4A(RNAi)* animals. The dorsal-specific expression of *bmp4* was largely intact in *nr4A(RNAi)* animals, with an increase in its expression in the esophagus compared with its expression in control animals (*Figure 3—figure supplement 3B*). There was no ectopic ventral *bmp4* expression in *nr4A(RNAi)* animals (*Figure 3—figure supplement 3B*). The midline expression domains of *slit* and *ephR1* were broadened medial-laterally and were disorganized in the head of *nr4A(RNAi)* animals (*Figure 3A*). *slit* expression was also reduced in the tail of *nr4A(RNAi)* animals, but its midline expression in the rest of the body was comparable to that of control animals (*Figure 3—figure supplement 3C*). This suggests *nr4A* inhibition does not cause a gross midline-patterning defect, but rather disrupts *slit* expression specifically in the head and tail, likely associated with disruption in the pattern of anterior and posterior poles, which play important roles in midline specification and patterning (*Hayashi et al., 2011*; *Currie and Pearson, 2013*; *März et al., 2013*; *Scimone et al., 2014*; *Vásquez-Doorman and Petersen, 2014*; *Vogg et al., 2014*; *Oderberg et al., 2017*).

Taken together, our PCG expression analysis showed that *nr4A* inhibition causes anterior PCGs to shift posteriorly and posterior PCGs to shift anteriorly (*Figure 3D*) while largely sparing midbody, midline, and DV PCG expression domains outside of the head and tail ends.

## *nr4A* is expressed in muscle and is upregulated at both anterior- and posterior-facing wounds when pattern begins to be reestablished in regeneration

Planarian muscle cells have important roles in expressing positional information to control adult body pattern maintenance and regeneration (*Witchley et al., 2013*; *Scimone et al., 2016*; *Scimone et al., 2017*). Consistent with its role in patterning, *nr4A* was predominantly expressed in planarian muscle in both intact animals and regenerating blastemas (*Figure 4A*). A higher proportion of muscle cells expressed *nr4A* in the head and tail than in the pre-pharyngeal region (*Figure 4B*). Although its expression was enriched at the ends of the AP axis, including in the posterior pole (*Figure 4C*), *nr4A* was also expressed broadly in body-wall muscle (*Figure 4A*). Mapping *nr4A* expression onto single-cell expression data (*Wurtzel et al., 2015*) showed that *nr4A* was most highly expressed in muscle cells, with some expression in neoblasts (*Figure 4D*). By using a single-cell sequencing dataset of different muscle cell subtypes (*Scimone et al., 2018*), we found that *nr4A* was expressed in longitudinal, circular, and DV muscle fiber types, but not in intestinal muscle cells (*Figure 4E*, *Figure 4—figure supplement 1*). Similarly, SCDE analysis of *nr4A*-expressing muscle cells using large-scale single-cell RNA sequencing data (*Fincher et al., 2018*) showed no enrichment for fiber-specific transcripts (*Supplementary file 1E*). During regeneration, *nr4A* expression was upregulated by 48 hpa and increased in level at 72 hpa at both anterior- and posterior-facing wound sites (*Figure 4F,G*). This expression time-course behavior in regeneration was similar to that of many anterior and posterior PCGs (*Wenemoser et al., 2012*; *Wurtzel et al., 2015*) (*Figure 4G*). However,

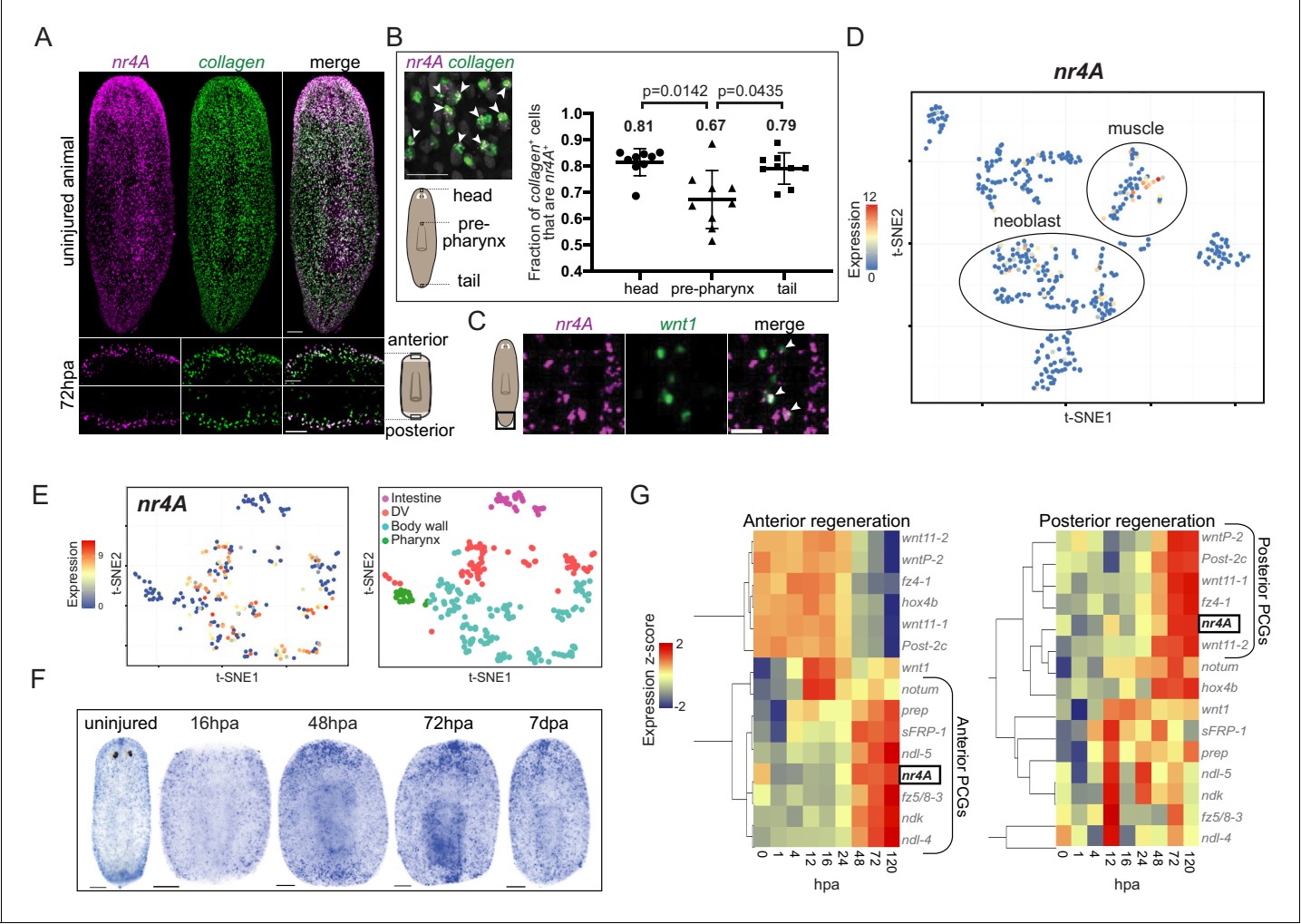

**Figure 4.** Planarian *nr4A* is expressed in muscle. (**A**) *nr4A* and *collagen* expression and co-expression in body-wall muscle cells in uninjured animals (top) and anterior and posterior bastema 72 hr post amputation (72hpa, bottom) by FISH. Blastema areas imaged are indicated by the boxes in the cartoon on the right. Uninjured image scale bar represents 100 μm. 72hpa image scale bar represents 50 μm. (**B**) Quantification of proportion of *collagen*[+] muscle cells that expressed *nr4A* by FISH in the head, pre-pharynx, and tail (*Figure 4—source data 1*), with a representative image showing *collagen* and *nr4A* double-positive cells (arrow heads). Numbers on the plot indicate mean proportions for each region. Statistical comparisons were done using Brown-Forsythe and Welch ANOVA tests with Dunnett's T3 multiple comparisons test. Scale bar represents 25 μm. (**C**) *nr4A* and *wnt1* expression and co-expression (arrow heads) in the posterior pole in uninjured animals by FISH. Area imaged is indicated by the box in the cartoon on the left. Scale bar represents 10 μm. (**D**) *nr4A* expression in single-cell clusters. Expression of *nr4A* mapped on the t-distributed Stochastic Neighbor Embedding (t-SNE) plot of cell type clusters generated by Seurat (*Wurtzel et al., 2015*). Muscle and neoblast clusters are labeled. (**E**) *nr4A* expression in muscle cell clusters (left) generated from single-cell RNA sequencing data in *Scimone et al. (2018)*; t-SNE representation of major muscle classes is on the right. (**F**) *nr4A* expression in uninjured animal and regenerating trunks at 16 hr post amputation (16hpa), 48hpa, 72 hpa, and 7 days post amputation (7dpa) by whole-mount in situ hybridization. All images are representative of results seen in at least four animals. (**G**) *nr4A* expression levels during anterior and posterior regeneration compared with PCG expression levels. Heat maps were generated with normalized expression levels (expression z-score) of *nr4A* and anterior and posterior PCGs during head (anterior) and tail (posterior) regeneration using previously published data (*Wurtzel et al., 2015*), and clustered by their trend of expression from 0 to 120 hr post amputation (hpa) with Spearman's correlation. *nr4A* clusters with anterior PCGs during anterior regeneration and posterior PCGs during posterior regeneration.

DOI: https://doi.org/10.7554/eLife.42015.012

The following source data and figure supplement are available for figure 4:

**Source data 1.** Quantification of proportion of *collagen*[+] muscle cells that express *nr4A* in head, pre-pharynx, and tail of uninjured animals.
DOI: https://doi.org/10.7554/eLife.42015.014

**Figure supplement 1.** Planarian *nr4A* is expressed broadly in body-wall muscle, including in longitudinal and circular muscle fibers.
DOI: https://doi.org/10.7554/eLife.42015.013

unlike canonical muscle-expressed patterning factors, *nr4A* was upregulated at both anterior- and posterior-facing wounds – having both anterior- and posterior-PCG-like expression dynamics during regeneration (*Figure 4G*).

## *nr4A* regulates the expression of muscle-specific markers and head and tail PCGs

Because *nr4A* encodes a transcription factor, we reasoned transcriptional changes in the head and tail tips of *nr4A(RNAi)* animals, such as changes in muscle-expressed patterning genes, might precede and explain the patterning phenotype observed. To identify such gene expression changes, we collected head and tail regions for RNA sequencing at several time points before gross anatomical changes could be detected: 2, 3, 4, and 5 weeks following the initiation of *nr4A* RNAi (*Figure 5A*).

We identified 56 and 41 genes with significant expression changes in the head and tail ($p_{adj}$ <0.05), respectively, in at least one time point in *nr4A(RNAi)* animals compared to controls (*Figure 5A*). The majority of these genes (49/56 in the head and 36/41 in the tail) were downregulated and most (32/56 in the head and 19/41 in the tail) were genes known to have muscle-enriched expression from available single-cell RNA sequencing data on wild-type animals (*Wurtzel et al., 2015*) (*Figure 5A*). 16 genes were significantly affected by *nr4A* in both the head and the tail. A high proportion (11/16) of those (e.g., dd_9565 (*col21a1*) and dd_11601 (*qki*)) were genes known to be expressed in muscle (*Wurtzel et al., 2015*). We verified the decrease in the expression of many of these genes following *nr4A* RNAi with FISH (*Figure 5B*, *Figure 5—figure supplement 1A*). Whereas *nr4A* RNAi reduced the body-wide expression of some genes (e.g., dd_9565 (*col21a1*), dd_1706 (*psapl1*)), it inhibited the expression of other genes more specifically in the head and tail (e. g., dd_2972, dd_11683 (*ca12*), dd_950 (*vim*)) (*Figure 5B*, *Figure 5—figure supplement 1A*). One of them, dd_950 (*vim*), was expressed in the epidermis (*Videos 1* and *2*). We also analyzed the expression patterns of genes with expression affected by *nr4A* RNAi using available RNA sequencing data from different AP axis segments (*Stückemann et al., 2017*) (*Figure 5—figure supplement 1B*). One-third of the *nr4A*-regulated genes, including 9 of 16 genes affected by *nr4A* in both the head and the tail, were enriched in their expression in the head and the tail compared to midbody regions (*Figure 5—figure supplement 1B*).

Among the genes with muscle-enriched expression affected by *nr4A* RNAi were several PCGs (*Figure 5A*). Following initiation of RNAi, *ndl-2* and *ndl-5* were downregulated in the head (by week 3 and 4, respectively), *wnt11-1* was upregulated in the tail (by week 2), and *nlg-8* was downregulated in the tail (by week 3). These findings are consistent with our FISH studies of the *nr4A(RNAi)* phenotype (*Figure 3A,C* and *Figure 5—figure supplement 1C*). These genes encode FGFRL (*ndl-2,–5*), Wnt (*wnt11-1*), and Bmp-regulatory (*nlg-8*) proteins and have been implicated in patterning (*Molina et al., 2007*; *Gurley et al., 2010*; *Lander and Petersen, 2016*; *Scimone et al., 2016*). These data show that *nr4A* is required for the expression of numerous muscle-expressed genes at the planarian head and tail ends, including those encoding extracellular matrix proteins and PCGs.

## Long-term *nr4A* RNAi causes head-specific reduction in muscle progenitor incorporation

Because a number of genes with muscle-enriched expression were downregulated following *nr4A* RNAi, we examined muscle fibers in these animals using the 6G10 muscle antibody (*Ross et al., 2015*). By 9 weeks of RNAi, when multiple ectopic eyes were present, uninjured *nr4A(RNAi)* animals showed loss of longitudinal, circular, and diagonal muscle fibers and a decreased number of collagen$^+$ muscle cells in the head compared to control animals (*Figure 6A–C*, *Figure 6—figure supplement 1A*). By contrast, muscle fibers were not significantly affected in the trunk or tail of *nr4A (RNAi)* animals and collagen$^+$ muscle cell numbers were normal in *nr4A(RNAi)* animal tails (*Figure 6A–C Figure 6—figure supplement 1A*). Similar analyses of *nr4A(RNAi)* animals at an earlier time point (4 weeks of RNAi) showed no changes in muscle fibers or collagen$^+$ cell numbers in the head or tail compared to control animals (*Figure 6A,C*, *Figure 6—figure supplement 1A–B*).

EdU labeling was performed at 9 weeks of *nr4A* RNAi to examine new muscle cell production. The numbers of new muscle cells (EdU$^+$; collagen$^+$) in the head tip of *nr4A(RNAi)* animals, but not in the posterior head, trunk, or tail, were significantly decreased compared to similar regions (by location and area) in control animals (*Figure 6D*). By contrast, there were no significant differences in the

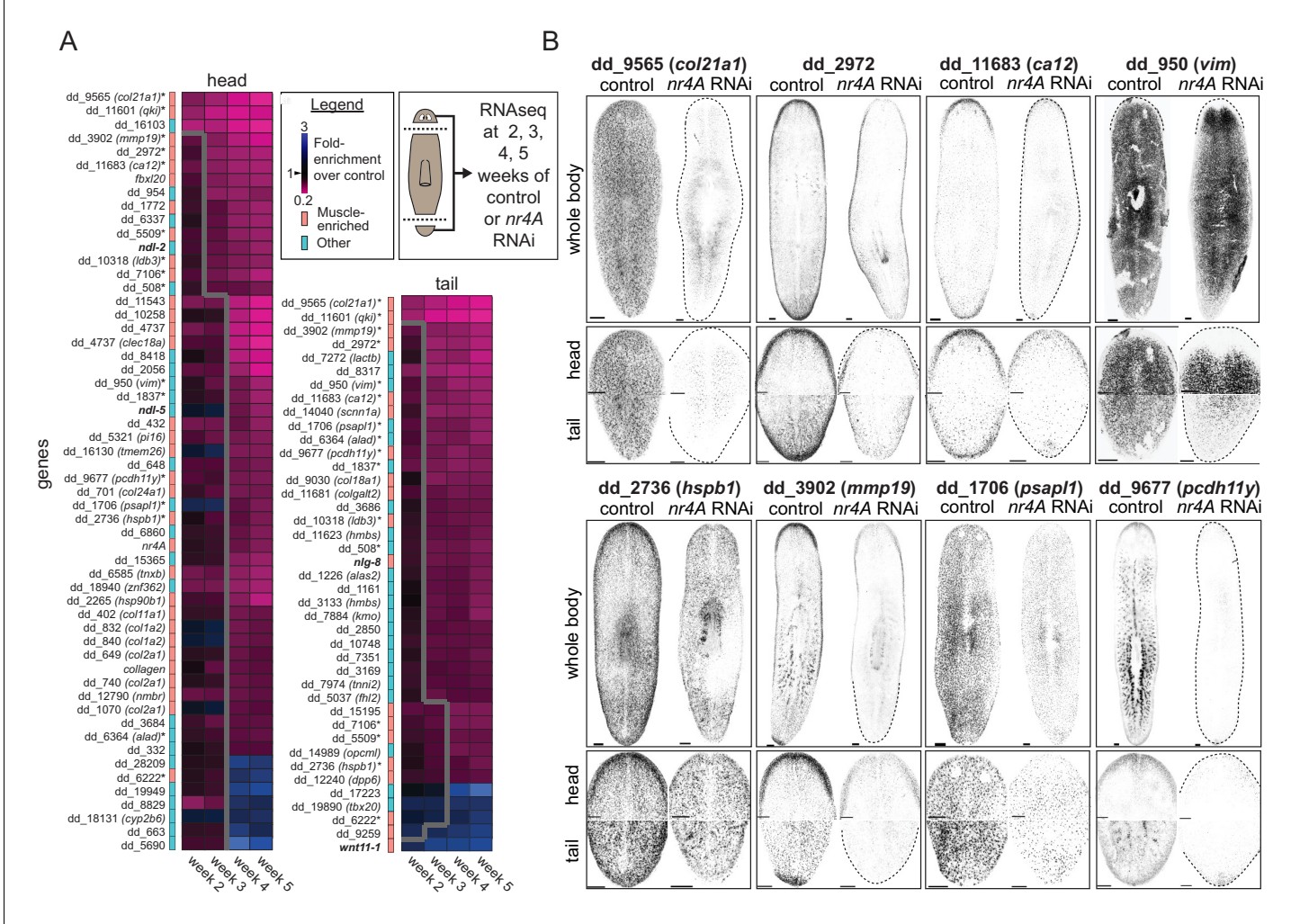

**Figure 5.** nr4A regulates the expression of muscle-enriched genes and PCGs. (**A**) RNA sequencing of control and nr4A(RNAi) animals: Diagram shows the collection of heads and tails for RNA sequencing at selected time points during the course of RNAi (see Materials and methods). Cartoon shows amputation planes as dotted lines. Heat maps were generated from the fold-enrichment of expression of genes differentially expressed in the heads or tails of nr4A(RNAi) versus control animals. Columns are RNAi time points and rows are individual genes. Heat map regions to the right of the thick gray line indicate statistically significant (p_adj <0.05) expression changes over control. Color-coded column to the left of each heat map indicates whether each gene was muscle-enriched (salmon) or not muscle-enriched (cyan), as previously determined (**Wurtzel et al., 2015**). Gene names in bold are PCGs; gene names with an asterisk are genes with significant expression changes in both the head and tail in nr4A RNAi compared to control. See Materials and methods for gene nomenclature. (**B**) Expression patterns of a subset of genes regulated by nr4A in both the head and the tail and their downregulation by nr4A RNAi, shown by FISH. All animals were analyzed at 9 weeks of RNAi. Dotted lines mark animal boundaries. Images are maximum intensity projections, and representative of results seen in at least three animals. Scale bars represent 150 μm.

DOI: https://doi.org/10.7554/eLife.42015.015

The following figure supplement is available for figure 5:

**Figure supplement 1.** nr4A-regulated gene expression patterns and regulation by nr4A RNAi.

DOI: https://doi.org/10.7554/eLife.42015.016

total amount of EdU incorporation (all EdU+ cells) in those regions between nr4A(RNAi) and control animals (**Figure 6—figure supplement 1C**). Comparing the region anterior to the original eyes in nr4A(RNAi) animals to the region anterior to the eyes in control animals showed greater reduction in new muscle cell number and reduction in total EdU incorporation (**Figure 6E**, **Figure 6—figure supplement 1D**). This finding is consistent with the head tip (distance between head apex and original eyes) size reduction observed in nr4A(RNAi) animals (**Figure 2A,C**). By contrast, the numbers of new muscle cells anterior to the eyes of controls and anterior to the newest eyes in nr4A(RNAi) animals

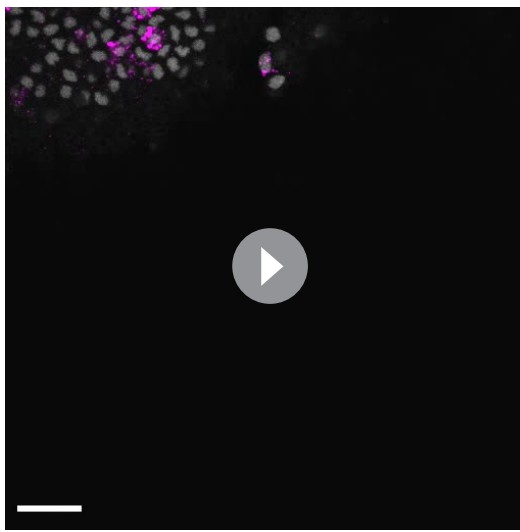

**Video 1.** Magnified image stack showing dd950(*vim*) expression (magenta) and epidermal nuclei (DAPI, gray) in control animal head by FISH.
DOI: https://doi.org/10.7554/eLife.42015.017

were similar (*Figure 6E*). *nr4A(RNAi)* and control animals had comparable numbers of apoptotic cells in similar head tip regions (by location and area) as assessed by TUNEL (*Figure 6—figure supplement 1E*), suggesting defective progenitor migratory targeting as opposed to increased cell death underlies muscle loss at the head tip.

These findings indicate that *nr4A* was not required in general for muscle progenitor production and differentiation throughout the body. The head-tip-specific muscle progenitor incorporation defect occurred after PCG and other gene expression changes were detected in the *nr4A(RNAi)* RNA sequencing data. Together with the observation that muscle fiber loss was also head-restricted and a late-stage phenotype (absent at 4 weeks of RNAi), these findings suggest loss of muscle cells per se is not the cause of the PCG expression changes that occurred at earlier timepoints in the *nr4A* RNAi.

## Early PCG domain shifts in *nr4A* RNAi precede changes in differentiated tissues at the head and tail tips

Given that the RNA sequencing results suggest PCG expression shifts precede differentiated tissue pattern changes, we sought to characterize the sequence of PCG expression changes that might underlie the phenotype. Importantly, we found that PCG shifts occurred before changes were visible in differentiated tissues, including changes in head muscle (*Figure 7A–D*). Specifically, in the head, *ndl-2* and *ndl-5* expression domains shifted posteriorly relative to the eyes by 3 weeks of *nr4A* RNAi compared to control RNAi (*Figure 7A,B*, *Figure 7—figure supplement 1A,B*, *Supplementary file 1D*). At this time point, the numbers of muscle cells and the appearance of muscle fibers were normal in the heads of *nr4A(RNAi)* animals (*Figure 6A,C*, *Figure 6—figure supplement 1A, B*). Mis-positioned anterior pole cells were the earliest abnormalities detected, becoming scattered between the eyes by two weeks after initiation of RNAi (*Figure 7A*). This posterior shift of *notum* expression preceded detected changes in the expression of other head PCGs (*ndl-2*, *ndl-4*, *ndl-5*, *sFRP-1*), as well as changes in the distribution of marginal adhesive gland cells (*Figure 7A, B*, *Figure 7—figure supplement 1A,B*, *Supplementary file 1D*). Consistent with our findings that muscle fiber loss at the head tip occurs at late stages of the phenotype (after 4 weeks - *Figure 6A,B*, *Figure 6—figure supplement 1A,B*), we observed no apparent differences in head tip muscle fibers between control and *nr4A(RNAi)* animals after two weeks of RNAi when pole cells were already gone from this location (*Figure 7C*, *Figure 7—figure supplement 1C*).

In the tail, the posterior pole shifted anteriorly by three weeks after initiation of RNAi (*Figure 7D*, *Supplementary file 1D*). The

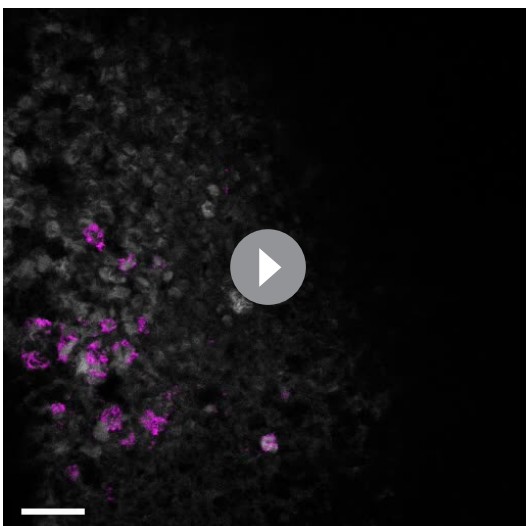

**Video 2.** Magnified image stack showing dd950(*vim*) expression (magenta) and epidermal nuclei (DAPI, gray) in *nr4A(RNAi)* animal head by FISH.
DOI: https://doi.org/10.7554/eLife.42015.018

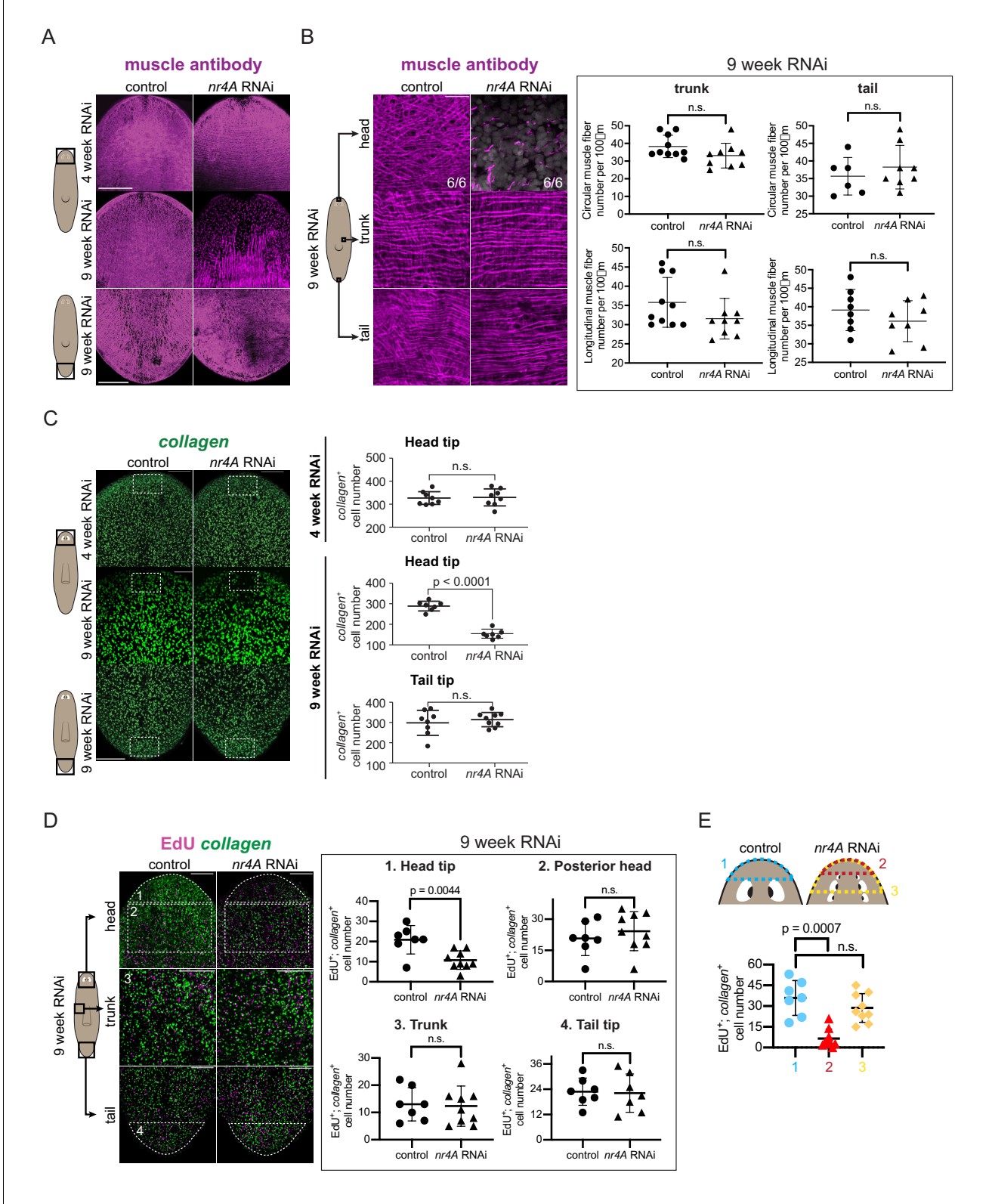

**Figure 6.** *nr4A* inhibition leads to muscle loss at the head tip. (**A**) Immunofluorescence with muscle antibody (6G10) in the head and tail of control and *nr4A(RNAi)* animals. Areas imaged are on the ventral surface of the animal. Scale bars represent 200 μm. (**B**) Higher magnification views of muscle fibers in the head tip, trunk, and tail of 9 week RNAi animals, and circular and longitudinal muscle fiber quantification in the tail and trunk (***Figure 6—source data 1***). Proportions in the head pictures indicate the number of animals with phenotype shown over total number of animals examined. Scale bars

*Figure 6 continued on next page*

*Figure 6 continued*

represent and 25 µm. Statistical comparisons were done using one-tailed Welch's *t*-tests. (C) Muscle marker *collagen* expression in the head and tail by FISH and quantification of *collagen*+ muscle cell number in a 190 µm x 125 µm area (white box) through the entire DV axis of head and tail tips (*Figure 6—source data 2*). Statistical comparisons were done using one-tailed Welch's t-tests. Scale bars represent 150 µm. (D) EdU labeling and *collagen* FISH of 9 week control and *nr4A(RNAi)* animals (left) and quantification of the number of new muscle (EdU+; *collagen*+) cells in each of the numbered regions (matched by location and area) between control and *nr4A(RNAi)* animals (right) (*Figure 6—source data 3*). Statistical comparisons were done using two-tailed Welch's *t*-test for cell counts in posterior head region and one-tailed Welch's *t*-tests for cell counts in all other regions. Scale bars represent 150 µm. (E) Quantification of the number of new muscle (EdU+; *collagen*+) cells anterior to the eyes in control animals (blue), and anterior to the original eye pair (red) and to the posterior-most ectopic eye pair (yellow) in *nr4A(RNAi)* animals (*Figure 6—source data 3*). Statistical comparisons were done using Brown-Forsythe and Welch ANOVA tests with Dunnett's T3 multiple comparisons test. n.s. = not significant at a level of 0.05 for all quantifications. All counting was performed blind. Areas imaged are indicated by the box in the cartoons on the left for all panels. All images are maximum intensity projections, and are representative of results seen in at least four animals per panel.

DOI: https://doi.org/10.7554/eLife.42015.019

The following source data and figure supplements are available for figure 6:

**Source data 1.** Muscle fiber quantifications at 9 weeks of RNAi.
DOI: https://doi.org/10.7554/eLife.42015.024
**Source data 2.** Quantification of number of *collagen*+ muscle cells at the head tip and tail tip of 9 week RNAi animals, and at the head tip of 4 week RNAi animals.
DOI: https://doi.org/10.7554/eLife.42015.025
**Source data 3.** EdU+; *collagen*+ quantifications in the head, trunk, and tail.
DOI: https://doi.org/10.7554/eLife.42015.026
**Figure supplement 1.** Effects of *nr4A* inhibition on muscle fibers, total EdU incorporation, and apoptosis.
DOI: https://doi.org/10.7554/eLife.42015.020
**Figure supplement 1—source data 1.** Muscle fiber quantifications at 4 weeks of RNAi.
DOI: https://doi.org/10.7554/eLife.42015.021
**Figure supplement 1—source data 2.** EdU+ cell quantifications in the head, trunk, and tail.
DOI: https://doi.org/10.7554/eLife.42015.022
**Figure supplement 1—source data 3.** TUNEL+ cell quantifications in the head.
DOI: https://doi.org/10.7554/eLife.42015.023

---

clustered expression of *wnt11-2* at the tail tip also decreased by this time point (*Figure 7D*, *Supplementary file 1D*). However, no changes in *mag-1*+ cells were apparent at this time (*Figure 7D*, *Supplementary file 1D*), indicating that similar to the case of the anterior head tip, PCG pattern changes precede differentiated tissue shifts at the posterior extremity of *nr4A(RNAi)* animals.

Even after just one week of *nr4A* RNAi, we observed a quantifiable increase in the distance between *notum*+ cells and the head apex (*Figure 8A*, *Figure 8—figure supplement 1A*, non-irradiated). Lethal irradiation kills the cycling cells (neoblasts) that are responsible for the production of all planarian cell types (*Bardeen and Baetjer, 1904*; *Dubois, 1949*; *Reddien et al., 2005*). Irradiation suppressed this earliest detectable posterior shift of *notum*+ cells in *nr4A(RNAi)* animals, demonstrating that anterior-pole changes were neoblast-dependent (*Figure 8A*, *Figure 8—figure supplement 1A* irradiated). SMEDWI-1 is a protein that persists in newly differentiated neoblast progeny (*Guo et al., 2006*). SMEDWI-1+ ectopic anterior pole cells were present at both early (2 weeks) and late (9 weeks) *nr4A* RNAi time points (*Figure 8B*), further indicating that new anterior pole cells are continually being targeted to ectopically posterior regions in *nr4A(RNAi)* animals. Relative to the original eyes, the anterior pole cells progressively appeared more posteriorly (*Figures 3A*, *7A* and *8A*).

## Anterior-pole shifting impacts head pattern alteration in *nr4A(RNAi)* animals

Evidence suggests that the anterior pole has organizer-like activity that can induce surrounding cells to adopt head tip identity (*Scimone et al., 2014*; *Vásquez-Doorman and Petersen, 2014*; *Vogg et al., 2014*; *Oderberg et al., 2017*). We hypothesized that rapid changes in anterior-pole-cell distribution following *nr4A* inhibition could be an important component in the development of the *nr4A(RNAi)* patterning phenotype. New anterior pole cell formation requires the gene *foxD*

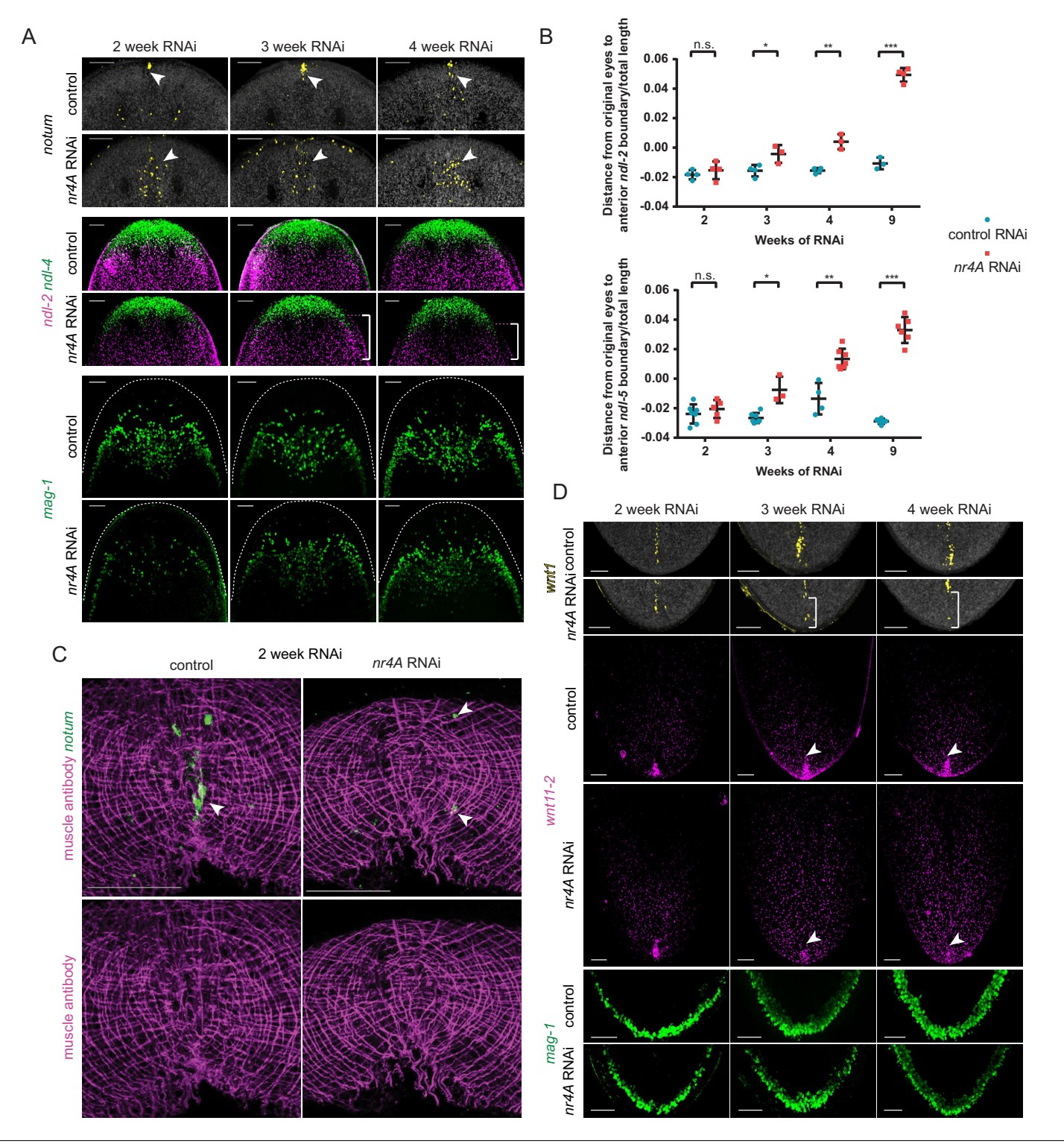

**Figure 7.** Changes in expression of head and tail PCGs precede changes in differentiated tissues in *nr4A* RNAi. (**A**) Head PCG expression by FISH in animals at 2, 3, and 4 weeks of RNAi. Anterior pole cells are indicated by arrow heads. Posterior shifts of the anterior boundary of the *ndl-2* expression domain are indicated by brackets. Quantification is shown in ***Supplementary file 1D***. Scale bars represent 100 μm. (**B**) Quantification of *ndl-2* (top) and *ndl-5* (bottom) expression domain shifts in a *nr4A(RNAi)* time course. Statistical comparisons were done using one-tailed Welch's *t*-tests. Negative and positive distances on *y*-axis denote that the position of the anterior boundary of expression domain is anterior and posterior to the eyes, respectively; n. s. = not significant at a level of 0.05. *=p < 0.05, **=p < 0.01, ***=p < 0.001. (**C**) Head-on view of head tips stained with muscle antibody 6G10 and

*Figure 7 continued on next page*

*Figure 7 continued*

*notum* probe at 2 weeks of RNAi. Anterior pole cells are indicated by arrow heads. Scale bars represent 50 μm. (**D**) Tail PCG expression by FISH in animals at 2, 3, and 4 weeks of RNAi. Anterior shifts in the posterior-most domain of *wnt1*+ posterior pole and the clustered expression domain of *wnt11-2* at tail tip are indicated by brackets and arrow heads, respectively. Quantification is shown in ***Supplementary file 1D***. Scale bars represent 100 μm. Images are maximum intensity projections. All images are representative of results seen in at least four animals per panel.

DOI: https://doi.org/10.7554/eLife.42015.027

The following figure supplement is available for figure 7:

**Figure supplement 1.** Shifts in PCG expression domains precede muscle fiber changes in *nr4A* RNAi.

DOI: https://doi.org/10.7554/eLife.42015.028

(***Scimone et al., 2014***; ***Vogg et al., 2014***), providing a method to genetically ablate the anterior pole. Uninjured animals were subjected to either *foxD* or control RNAi for 6 weeks to eliminate the anterior pole (a subgroup, "pole-cut animals", was subjected to head tip excision to facilitate anterior pole removal). Subsequently, *nr4A* RNAi was added to *foxD* and control(RNAi) groups (10 weeks of *nr4A* RNAi for intact animals and 12 weeks for pole-cut animals) to assess whether ectopic pole cell placement was required for the *nr4A(RNAi)* phenotype. A greater proportion of *foxD*; *nr4A (RNAi)* animals lacked ectopic eyes than did control; *nr4A(RNAi)* animals (***Figure 8C***). This result was consistently observed in intact and pole-cut animal groups, and in biological replicates (***Figure 8— figure supplement 1B,C***). qPCR of animals at the end of the double RNAi period showed similar levels of *nr4A* expression inhibition between *foxD*; *nr4A(RNAi)* and control; *nr4A(RNAi)* groups (***Figure 8D***, ***Figure 8—figure supplement 1D***). Taken together with the fact that a posterior shift in anterior pole cells was the earliest observed change in *nr4A(RNAi)* animals, these results indicate that the head patterning defects of *nr4A* RNAi are mediated, at least in part, by the mis-positioning of anterior pole cells.

### Head tissue pattern fails to reach a stable state in *nr4A(RNAi)* animals

The iterative appearance of ectopic eyes in *nr4A(RNAi)* animals prompted us to ask if a stable, but different, tissue pattern ever emerged in these animals. Tracking the fates of the eyes in individual animals after extended periods of continuous *nr4A* inhibition (as long as 21 weeks) showed that the original eyes (and even older ectopic eyes) eventually faded and disappeared at the head tip as the phenotype progressed (***Figure 9A***, ***Figure 9—figure supplement 1***). As this process unfolded, eye progenitors iteratively shifted their targeting to the most posterior set of eyes, leaving the anterior eyes to decay. Specifically, there were significantly more new eye cells (*opsin*+; SMEDWI+) in the newest (posterior) ectopic eyes than in older (more anterior) ectopic eyes (***Figure 9B***). The numbers of new eye cells produced in a given unit of time were similar between the eyes of control animals and the newest, posterior ectopic eyes in *nr4A(RNAi)* animals (***Figure 9B***). These findings suggest that eye progenitor targeting continued to shift posteriorly after *nr4A* inhibition, iteratively producing ectopic eyes without reaching a stable tissue-pattern state.

## Discussion

### Anterior- and posterior-pole transcriptomes

The planarian anterior and posterior poles have been the subjects of recent intense study because of their roles in patterning the head and tail (***Reddien, 2011***; ***Owlarn and Bartscherer, 2016***; ***Reddien, 2018***). The poles are specialized muscle cells localized at the midline and at the extreme ends of the AP axis and are specified by transcription factors as discrete structures at the animal ends. Given the role for poles in influencing the pattern of neighboring tissues, the poles have been regarded as organizers in planarian regeneration (***Hayashi et al., 2011***; ***Currie and Pearson, 2013***; ***März et al., 2013***; ***Scimone et al., 2014***; ***Vásquez-Doorman and Petersen, 2014***; ***Vogg et al., 2014***; ***Vogg et al., 2016***).

Using bulk and single-cell RNA sequencing, we identified many new genes expressed in the anterior and posterior poles, including genes expressed in both poles that suggest functional similarities in these cells. For example, dd_13188 (*kallmann1*) is expressed highly specifically in both poles and encodes an extracellular matrix protein implicated in olfactory neuron migration in humans

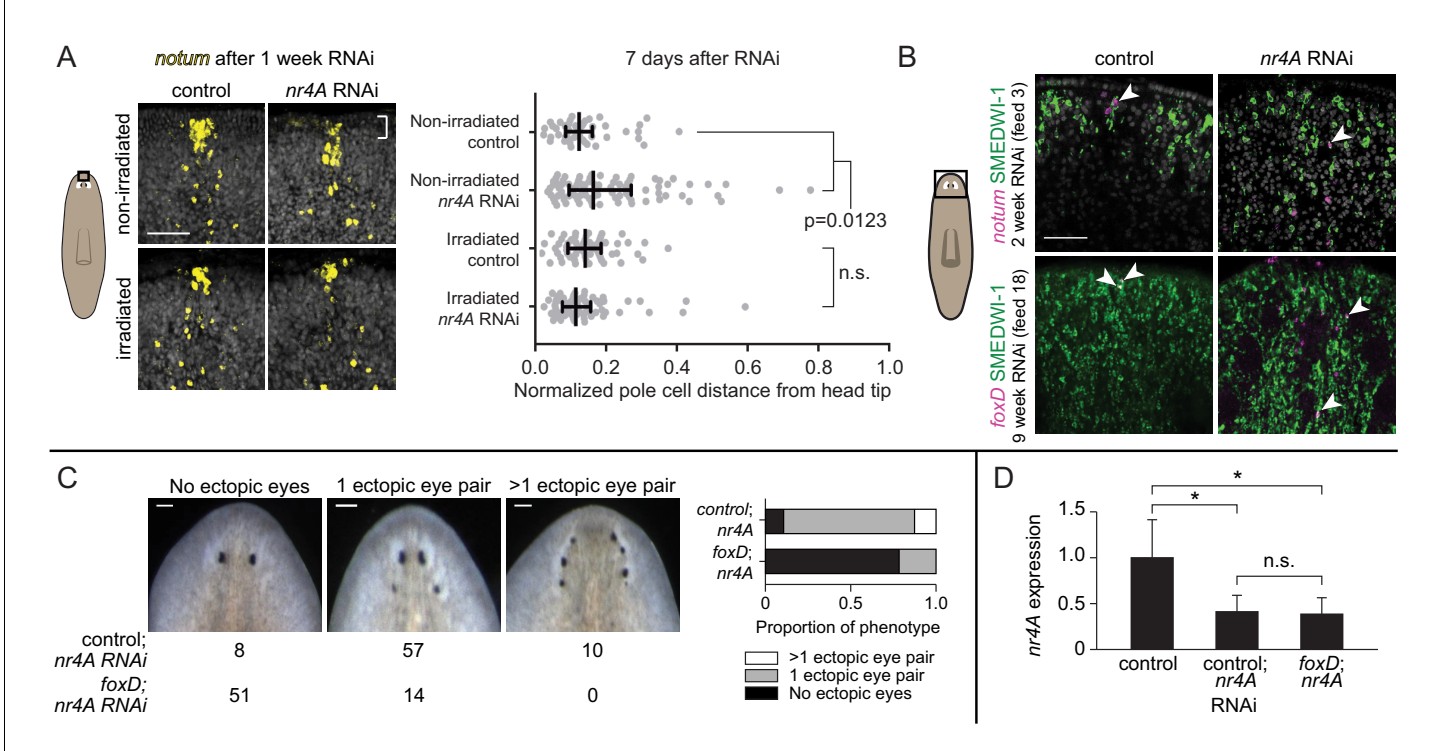

**Figure 8.** Anterior pole cells shift early and contribute to the head patterning phenotype in *nr4A(RNAi)* animals. (**A**) Expression of *notum* by FISH in animals fixed 7 days after a single *nr4A* or control(RNAi) feeding and 8 days after lethal gamma irradiation. Distances of individual anterior pole cells to head apex were normalized to distances between the eyes within each animal (*Figure 8—source data 1*). Statistical comparisons were done with Kruskal-Wallis test with Dunn's multiple comparisons test. Scale bar represents 50 μm. (**B**) FISH with *notum* or *foxD* probe and SMEDWI-1-antibody labeling in the head after 2 and 9 weeks of RNAi. Arrow heads indicate *notum*[+]; SMEDWI-1[+] or *foxD*[+]; SMEDWI-1[+] cells. Scale bar represents 50 μm. (**C**) *foxD; nr4A* double RNAi phenotype scoring. Representative live images of animals within each phenotypic category (no ectopic eyes, one ectopic eye pair, and more than one ectopic eye pair) are shown. Numbers of animals with each phenotype in control; *nr4A* and *foxD; nr4A* RNAi are displayed below, with graphic representation of phenotype proportions on the right. Differences in the number of animals seen in each phenotypic category between control; *nr4A* and *foxD; nr4A* RNAi were statistically significant (chi-squared = 67, p<0.00001, df = 2). Scale bars represent 150 μm. (**D**) qPCR quantification of *nr4A* expression in control, control; *nr4A*, and *foxD; nr4A* RNAi in uninjured animals. Statistical comparisons were done with Brown-Forsythe and Welch ANOVA tests with Dunnett's T3 multiple comparisons test. * = p<0.05. n.s. = not significant at a level of 0.05 for all quantifications. Area imaged is indicated by the box in the cartoon on the left for panels A and B. All images are representative of results seen in at least four animals per panel.

DOI: https://doi.org/10.7554/eLife.42015.029

The following source data and figure supplement are available for figure 8:

**Source data 1.** Quantification of anterior pole cell distance from head apex, normalized to inter-eye distance in each animal.
DOI: https://doi.org/10.7554/eLife.42015.031

**Figure supplement 1.** Anterior pole shift, a neoblast-dependent process, is important for ectopic eye formation in *nr4A* RNAi.
DOI: https://doi.org/10.7554/eLife.42015.030

(*Rugarli et al., 1993*; *Rugarli, 1999*). Genes with unique functions in organizers in development and regeneration are of interest for understanding how these regions form and pattern neighboring tissue, and RNA-sequencing approaches like the one defined here could identify the transcriptomes of these regulatory regions of embryos and animals broadly.

## *nr4A* is a muscle-expressed transcription factor required for patterning

Planarian *nr4A* is specifically expressed in muscle and is required for AP axial patterning. *nr4A* genes are widely conserved in the animal kingdom, and to our knowledge this is the first report of a role in tissue patterning for a member of the NR4A-family of nuclear receptors. The most striking aspect of the activity of *nr4A* in planarians is its role in homeostatic patterning at both ends of the AP axis. Patterning phenotypes affecting both ends of the AP axis of animals are rare. In planarians, *islet1* is

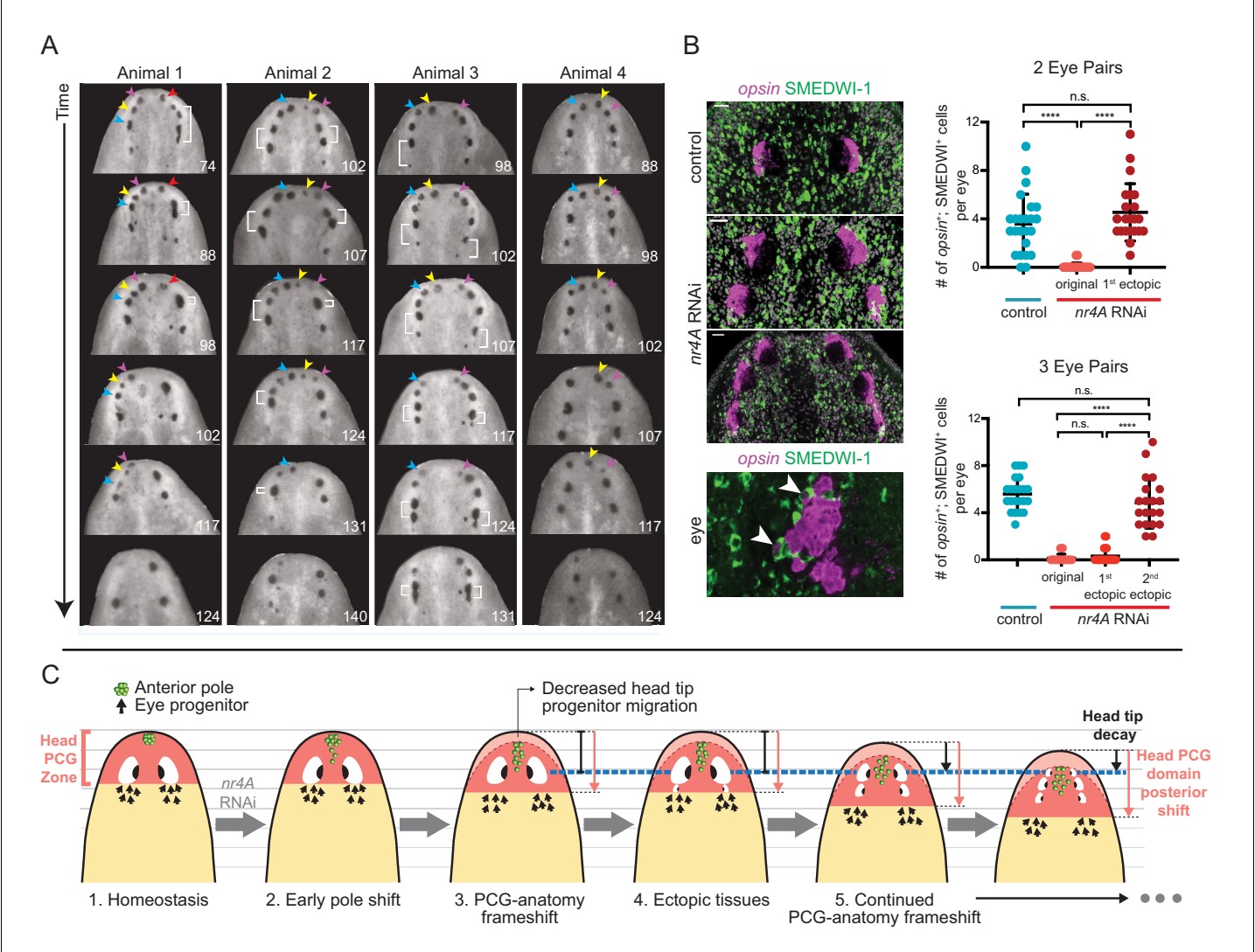

**Figure 9.** *nr4A* RNAi leads to patterning that fails to reach a stable state. (**A**) Disappearance and fusion of eyes after many weeks of *nr4A* RNAi. Single animals were separated and individually imaged over the course of long-term *nr4A* RNAi. The number of days of RNAi at the time image was taken is at the bottom right of each image. Eyes crowded near the head tip in *nr4A(RNAi)* animals were observed to disappear (colored arrow heads in frames in which they appear) or fuse (white brackets). (**B**) Progenitor incorporation into the eyes of RNAi animals. Newly incorporated (*opsin*+; SMEDWI-1+) cells in each eye were quantified in control, 2-eye-pair *nr4A(RNAi)* animals, and 3-eye-pair *nr4A(RNAi)* animals (*Figure 9—source data 1*), with representative images of the eyes in each group of animals shown. A magnified image of an eye with *opsin*+; SMEDWI-1+ cells (arrow heads) is shown at the bottom. Double-positive cells were included in the count if located within the eye itself or within approximately one cell diameter of the eye. Statistical comparisons were done with Brown-Forsythe and Welch ANOVA tests with Dunnett's T3 multiple comparisons test. n.s. = not significant at a level of 0.05. **** = p<0.0001. (**C**) *nr4A(RNAi)* phenotype model (animals aligned by the position of the original eyes, marked by the blue dotted line): *nr4A* RNAi causes an early posterior shift in anterior pole (green) cells followed by a similar shift in other head PCG domains (red zone), and formation of new tissues at ectopically posterior positions with respect to the original anatomy (e.g., posterior ectopic eyes). As PCG domains continue to shift in relation to anatomy, this process continues iteratively throughout *nr4A* inhibition. Refer to Discussion for a more detailed description.

DOI: https://doi.org/10.7554/eLife.42015.032

The following source data and figure supplement are available for figure 9:

**Source data 1.** SMEDWI-1+; *opsin*+ cell quantification in eyes of control and *nr4A(RNAi)* animals.

DOI: https://doi.org/10.7554/eLife.42015.034

**Figure supplement 1.** Anterior eyes disappear and fuse in long-term *nr4A* RNAi.

DOI: https://doi.org/10.7554/eLife.42015.033

known to be required for both anterior and posterior pole formation (*Hayashi et al., 2011*; *März et al., 2013*). Here, we find a unique patterning phenotype affecting pole positioning and differentiated tissue pattern at both extreme ends of the AP axis.

NR4A-subfamily members belong to the broader superfamily of nuclear receptors specific to metazoans (*Bridgham et al., 2010*; *Escriva et al., 2004*). Although members of many other subfamilies of nuclear receptors bind to steroid hormones, retinoic acids, fatty acids, and prostaglandins, NR4A-family members are orphan receptors that have been proposed to regulate transcription in a ligand-independent manner (*Paulsen et al., 1992*; *Bridgham et al., 2010*). Whereas vertebrates possess three NR4A members (NR4A1, NR4A2, and NR4A3), many protostomes have only one (*Bertrand et al., 2004*). DHR38, the *Drosophila* NR4A ortholog, mediates an atypical ligand-independent ecdysteroid-signaling pathway responsible for molting and metamorphosis (*Baker et al., 2003*). In vertebrates, three members of the *NR4A* family are immediate-early stress response genes induced by a host of physiological signals such as growth factors, cytokines, prostaglandins, neurotransmitters, and phorbol esters (*Maxwell and Muscat, 2006*; *Safe et al., 2016*). NR4A1, NR4A2, and *NR4A3* are expressed in a variety of tissues, such as the brain, liver, gonads, skeletal muscle, kidney, fat, lung, and endocrine glands (*Maxwell and Muscat, 2006*). In the fields of developmental biology and regeneration, little is known about the role of NR4A transcription factors. *nr4A* was expressed strongly in the planarian head and tail, but was also expressed in muscle cells throughout the body. After amputation, *nr4A* was upregulated at anterior-facing wounds, with timing similar to anterior PCGs, and at posterior-facing wounds, with timing similar to posterior PCGs. RNA-sequencing data from *nr4A(RNAi)* animals revealed that many of the genes dependent on *nr4A* for their expression were specifically expressed in muscle cells, the majority of which encode extracellular matrix (ECM) components (collagens and metalloproteinase) and signaling proteins. Several of these genes predicted to encode ECM components displayed enriched expression in both the head and tail tips. Although it is unknown which genes are direct transcriptional targets of the NR4A protein, genes that experienced changes in expression early in the course of RNAi are more likely to be directly dependent on *nr4A*.

## A phenotype that fails to reach a stable pattern

The planarian AP axis is associated with a continuum of distinct and overlapping PCG expression domains (*Forsthoefel and Newmark, 2009*; *Adell et al., 2010*; *Reddien, 2011*). PCGs promote the production of varying cell types and tissue patterns along the AP axis, for example by promoting eye formation in the head and pharynx formation in the midbody (*Lander and Petersen, 2016*; *Scimone et al., 2016*). Progenitors for differentiated tissues are specified by the expression of TF genes in neoblasts, generating specialized neoblasts (*Reddien, 2013*; *Zhu and Pearson, 2016*). These specialized neoblasts are specified coarsely on the AP axis and produce progenitors that migrate to specific locations or "target zones" (TZs) (*Atabay et al., 2018*). For example, eye-specialized neoblasts are specified broadly in the head and pre-pharyngeal regions and migrate to precise locations in the head where eyes form and are maintained (*Lapan and Reddien, 2011*; *Lapan and Reddien, 2012*). Altering PCG pattern with RNAi can result in mis-targeting of progenitors. For example, inhibition of the laterally expressed *wnt5* gene causes eye progenitors to be targeted more laterally than in the wild type, resulting in ectopic lateral eyes (*Atabay et al., 2018*). In addition to migratory targeting of progenitors by extrinsic cues, self-organization of progenitors into their target organ/tissue has a major influence on progenitor targeting (*Atabay et al., 2018*). For example, in instances when an organ and the PCG pattern are discordant (such as during morphallaxis), progenitors can be incorporated into such an organ despite the fact that it is in the incorrect position (*Atabay et al., 2018*; *Hill and Petersen, 2018*).

We propose a model for the *nr4A(RNAi)* phenotype based upon this prior knowledge and the work presented here. The model involves ectopically posterior tissues in the head and ectopically anterior tissues in the tail manifesting from shifts of the AP positional information away from AP axis ends. This model is described in *Figure 9C* and below: Focusing on the head, the AP PCG axis terminus demarcated by the anterior pole shifts internally from the AP anatomical terminus (the head tip) in *nr4A(RNAi)* animals. These pole cells influence gene expression in neighboring muscle so that the entire PCG pattern for the head shifts posteriorly. This results in an animal with a frameshift between the positional information map and the anatomy (*Figure 9C*). PCG expression shifting changes the target-zone locations for progenitors. For example, once the target zone for the eye

has shifted sufficiently posterior to avoid the self-organizing influence of the original eyes, ectopic posterior eyes appear (*Figure 9C*). At this point the original eyes stop receiving progenitors and shrink, as was observed. The head tip shrinks for similar reasons - without the PCG extremity zone coincident with the head tip, less progenitor targeting to the tip leads to its decay during natural tissue turnover (*Figure 9C*).

This patterning abnormality continues in an iterative process: new sets of posterior eyes continue to appear and anterior eye sets and the head tip shrink. What underlies this iterative process? We propose that the continuous out-of-register placement of pattern-organizing cells (new pole cells that are part of naturally occurring tissue turnover) in relation to the rest of positional information underlies this process (*Figure 9C*). Anterior pole progenitors would continue to be targeted too posteriorly in the changing PCG map. This continues to influence PCG pattern, shifting it even farther posteriorly. Consequently, the extremity continues to decay and the target zone for eye progenitors continues to move posteriorly, with sets of eyes iteratively appearing posteriorly, and anterior sets of eyes decaying because of a lack of progenitor targeting for their renewal. Tracking new eye progenitor incorporation supported this model (*Figure 9B*). This process would continuously occur, with equilibrium between PCG pattern and anatomy pattern not attainable. This novel type of patterning phenotype suggests that breaking the concordance between new pattern-organizing cells and pattern itself can lead to a phenotype of continuous pattern-anatomy shifting.

This model depends on PCG-pattern shifts preceding differentiated tissue changes, as was observed. As early as one week after *nr4A* RNAi, the anterior pole began to shift away from the head tip, followed by the posterior pole shifting away from the tail tip at 3 weeks of RNAi. Between 3 and 4 weeks of RNAi, a host of other head and tail PCGs became progressively excluded from head and tail tips. Ectopic eyes, gland, or DV-boundary epidermal cells all appeared later, between 6 and 12 weeks of RNAi. This model is also supported by the finding that ectopic anterior pole cells in *nr4A(RNAi)* animals were newly specified from neoblasts and were required for the *nr4A(RNAi)* phenotype. Ultimately, muscle progenitor targeting to the head tip was reduced in *nr4A(RNAi)* animals, but this occurred after pole and other PCG pattern shifts initiated. This reduction could reflect a general defect in head tip targeting of muscle and other progenitors, perhaps first manifested by the shift in pole progenitor placement, which triggered pattern changes in head muscle.

The *nr4A(RNAi)* phenotype is distinct from prior RNAi phenotypes affecting the number of planarians eyes or pattern at the ends of the AP axis. Unlike *notum* RNAi in uninjured animals, which results in anterior ectopic eyes (*Atabay et al., 2018*; *Hill and Petersen, 2018*), *nr4A* RNAi generates posterior ectopic eyes. In *ndk* RNAi, ectopic eyes appear in very posterior regions (including the trunk and tail) (*Cebrià et al., 2002*), whereas in *nr4A* RNAi, ectopic posterior eyes are restricted to the head. Furthermore, *ndk* RNAi does not lead to a posterior shift in the anterior pole (*Scimone et al., 2016*). Finally, although *islet1* inhibition also affects both poles, its patterning defect during regeneration (versus during homeostasis for *nr4A* RNAi) leads to midline collapse (e.g., cyclopia) (*Hayashi et al., 2011*; *März et al., 2013*) that is not observed in *nr4A(RNAi)* animals.

Why would the iterative tissue treadmill-like process of the *nr4A(RNAi)* phenotype stay regionally restricted, without progressing through the body? Whereas PCG expression in the head tip retracted posteriorly, expression boundaries of PCGs outside of the head (e.g., the posterior expression boundary of *ndl-2, ndl-5,* and *wnt2*) did not move posteriorly. Therefore, if head progenitors in general are still made at normal rates (as seen for eye progenitors), one possibility is that more progenitor targeting towards the posterior head could result in growth counterbalancing head tip shrinking.

Our findings regarding the *nr4A(RNAi)* phenotype are consistent with and provide additional evidence for the presence of PCG-defined progenitor target zones that both maintain proper tissue structures during homeostasis and instruct tissue formation at new locations as PCG zones shift, for example, in regeneration (*Atabay et al., 2018*; *Hill and Petersen, 2018*). Together, our data show that *nr4A* is necessary to maintain the correct location of both the anterior- and posterior-PCG expression zones, consequently maintaining the proper pattern of positional information and differentiated tissues in the head and the tail. We conclude that *nr4A* is a novel adult patterning gene that helps muscle promote the patterning of both the anterior and posterior extreme ends of the primary planarian body axis.

# Materials and methods

**Key resources table**

| Reagent type (species) or resource | Designation | Source or reference | Identifiers | Additional information |
|---|---|---|---|---|
| Gene (*Schmidtea mediterranea*) | *nr4A*; dd_9565 (*col21a1*); dd_3902 (*mmp19*); dd_2972; dd_11683 (*ca12*); dd_508; dd_950 (*vim*); dd_9677 (*pcdh11y*); dd_1706 (*psapl1*); dd_2736 (*hspb1*); dd_11601 (*qki*) | This paper | RRID: NCBITaxon:79327 | New *Schmidtea mediterranea* genes (*nr4A* and *nr4A*-regulated genes) cloned in this paper. See **Supplementary file 1F** for primer sequences. |
| Strain, strain background (ClW4 clonal line) | Asexual strain of *Schmidtea mediterranea* | NA | RRID: NCBITaxon:79327 | All animals used in this study |
| Antibody | rabbit anti-SMEDWI-1 polyclonal antibody | *Guo et al., 2006* | RRID: AB_2797418 | 1:1000 dilution |
| Antibody | anti-muscle mouse monoclonal 6G10 | *Ross et al., 2015* (DSHB Hybridoma Product 6G10-2C7) | RRID: AB_2619613 | 1:1000 dilution, antibody used for muscle fiber staining |
| Commercial assay or kit | TruSeq RNA Library Prep Kit V2 | Illumina | RS-122–2001, RS-122–2002 | Used for bulk RNA sequencing |
| Commercial assay or kit | Nextera XT DNA Library Preparation Kit | Illumina | FC-131–1002 | Used for single-cell RNA sequencing DNA library construction |
| Commercial assay or kit | ApopTag Red In Situ Apoptosis Detection Kit | Millipore | S7165 | Used for TUNEL staining |
| Chemical compound, drug | F-ara-EdU | Sigma-Aldrich | T511293 | Used for EdU labeling |
| Software, algorithm | Prism 8 | GraphPad | RRID: SCR_002798 | Used for all statistical analyses |
| Software, algorithm | Fiji/ImageJ | NIH, public | RRID: SCR_002285 | Used for all image contrast adjustment and cell, muscle fiber, and PCG domain quantification |
| Software, algorithm | SCDE | *Kharchenko et al., 2014* | RRID: SCR_015952 | Used in single-cell RNA sequencing differential expression analysis |
| Software, algorithm | DESeq | *Anders and Huber, 2010* | RRID: SCR_000154 | Used in bulk RNA sequencing differential expression analysis |
| Software, algorithm | MacVector | MacVector, Inc | RRID: SCR_015700 | Used to align protein sequences |
| Software, algorithm | ClustalX | Clustal | RRID: SCR_017055 | Used to create .phy protein sequence alignment files |

## Animals

Asexual strain ClW4 of *Schmidtea mediterranea* (RRID: NCBITaxon:79327) was used for all experiments. Animals were starved 7–14 days before experiments.

## Bulk RNA sequencing

Cut animal fragments were diced with a scalpel, homogenized using Qiagen TissueLyser III, and RNA was extracted using TRIzol (Life Technologies) according to the manufacturer's protocol. Between 0.5 and 1 µg of RNA was used for cDNA sequencing library synthesis using TruSeq RNA

Library Prep Kit V2 (Illumina) following the manufacturer's protocol. Libraries were sequenced on Illumina HiSeq for an average sequencing depth of 20–30 million reads per replicate sample. Libraries were mapped to the dd_Smed_v4 transcriptome (http://planmine.mpi-cbg.de) using bowtie 1 (*Langmead et al., 2009*) with -best alignment parameter. Pairwise differential expression analysis was performed using DESeq (*Anders and Huber, 2010*) (RRID: SCR_000154). Sequencing data and DESeq analysis available in NCBI/GEO (accession GSE121048).

## Single-cell RNA sequencing and expression analysis

Cut animal fragments were diced with a scalpel, macerated with collagenase treatment, stained with Hoechst (1:25) and propidium iodide (1:5000) and 2C (differentiated, non-dividing) viable (Hoechst$^+$; propidium iodide$^-$) cells were sorted (*Hayashi et al., 2006*) one cell per well by fluorescence activated cell sorting on 96-well plates containing 5 µL of Buffer TCL with 1% 2-mercaptoethanol. Single-cell RNA sequencing libraries were prepared via the SmartSeq2 method, as previously described (*Picelli et al., 2013*; *Picelli et al., 2014*). Briefly, a poly-dT oligo and a template-switching oligo were used for reverse transcription. After cDNA amplification, single-cell cDNA libraries were fragmented and tagged using the Nextera XT kit (Illumina). Single-cell cDNA libraries were then qRT-PCR screened for *wnt1* and *collagen* expression with the primers provided in *Supplementary file 1F*. Posterior pole cells with *wnt1* expression (11) and muscle cells with *collagen* but no *wnt1* expression (90) were selected for were sequencing on Illumina HiSeq for an average of 1–2 million reads per cell. Differential expression analysis of gene expression between pole cells and non-pole cells was performed using the Single-Cell Differential Expression (SCDE, RRID: SCR_015952) method, as described (*Kharchenko et al., 2014*). Sequencing data and SCDE analysis available in NCBI/GEO (accession GSE121048).

## Gene nomenclature

Previously published and homology-verified planarian genes are in lowercase italics by themselves. Genes not analyzed for homology by phylogenetics but that have strong (E-value <0.05) human best-BLAST matches have gene identifiers with "dd_" followed by their Smedv4.1 Dresden transcriptome assembly contig number and italicized human best-BLAST match in parentheses. Genes that only have Dresden transcriptome assembly contig identifiers have no human, mouse, or *C. elegans* best BLAST matches.

## Phylogenetic analysis

Nuclear receptor family phylogenetic tree was constructed from 114 total protein sequences (including the translated *Schmidtea mediterranea* dd_12229 *nr4*A mRNA sequence) from representative nuclear receptors of all six nuclear receptor subfamilies from eight different species plus *Schmidtea mediterranea*. Their accession numbers are in *Supplementary file 1G*. Protein sequences were aligned by their conserved DNA-binding and ligand-binding domains via maximum likelihood method using PhyML with 1000 bootstrap replicates. Trees were visualized and formatted in FigTree.

## Gene cloning

Primers used to clone *nr4A* and its targets are listed in *Supplementary file 1F*. Genes were cloned from cDNA into the pGEM vector (Promega) and transformed into *E. coli* DH10B by heat shock. Bacteria were plated on agarose plates containing 1:500 carbenicillin, 1:200 Isopropylthio-β-D-galactoside (IPTG), and 1:625 5-bromo-4-chloro-3-indolyl-β-D-galactopyranoside (X-gal) for overnight growth. Colonies were screened by colony PCR and gel electrophoresis. Plasmids were extracted from positive (white) colonies and subsequently validated by Sanger sequencing.

Single RNAi dsRNA was transcribed in vitro (Promega reagents) from PCR-generated templates with flanking T7 promoters. It was then precipitated in ethanol, annealed after resuspension in water, and mixed with planarian food (liver). Each animal was fed 2 µL of the liver containing 4–7 µL/mL dsRNA twice every week, with 2–3 days between each feeding. Animals were then fixed seven days after the last feeding. The total amount of dsRNA per feeding per animal was kept constant as described before. Control RNAi used dsRNA against the *C. elegans unc-22* transcript.

## In situ Hybridizations

WISH with nitro blue tetrazolium/5-bromo-4-chloro-3-indolyl phosphate (NBT/BCIP) was performed as described (*Pearson et al., 2009*). FISH and post-antibody binding washes and tyramide development were performed as described (*King and Newmark, 2013*). Briefly, animals were killed in 5% NAC and treated with proteinase K (2 µg/ml). Animals were hybridized with RNA probes at 1:800 dilution overnight at 56˚C. Samples were then washed twice in pre-hybridization buffer, 1:1 pre-hybridization: 2X SSC, 2X SSC, 0.2X SSC, and PBS with Triton-X (PBST) in that sequence. Blocking was performed in 5% Western Blocking Reagent (Roche) plus 5% heat-inactivated horse serum diluted in PBST solution when anti-DIG, anti-FITC, or anti-DNP antibodies were used. Post-antibody binding washes and tyramide development were performed as described (*King and Newmark, 2013*). Light images were taken with a Zeiss Discovery Microscope. Fluorescent images were taken with a Zeiss LSM700 Confocal Microscope. Fiji/ImageJ (RRID: SCR_002285) was used for FISH co-localization analyses and PCG domain quantifications.

## Immunostaining

Animals were fixed in 4% formaldehyde solution as for in situ hybridization and treated as described (*King and Newmark, 2013*). The muscle antibody 6G10 (*Ross et al., 2015*) (RRID: AB_2619613) was used at 1:1000 dilution in PBSTB (0.1% TritonX, 0.1% BSA), and an anti-mouse-Alexa conjugated antibody (Life Technologies) was used at a 1:500 dilution. The rabbit anti-SMEDWI-antibody (RRID: AB_2797418) was used at 1:1000 dilution in PBSTx (0.1% TritonX) with 10% horse serum, and an anti-rabbit-HRP antibody was used at 1:300 dilution in PBSTx (0.1% TritonX) with 10% horse serum.

## Cell and expression domain quantification

All quantification was performed in a condition-blind manner. Statistical analyses were done using Welch's *t*-tests for comparisons between two groups or using Brown-Forsythe and Welch ANOVA tests with Dunnett's T3 multiple comparisons test for comparisons involving more than two groups, unless otherwise specified. For the proportion of muscle cells expressing *nr4A*, the number *collagen*[+]; *nr4A* [+] cells was divided by the number of *collagen*[+] cells in head tip, pre-pharyngeal region, and tail tip in a 200 µm by 200 µm area through a 15 µm stack (*Figure 4—source data 1*). The number circular and longitudinal muscle fibers in each animal was counted within a 100 µm x 100 µm image area (*Figure 6—source data 1*, *Figure 6—figure supplement 1—source data 1*). For muscle cell quantification, *collagen*[+] cells were counted in a in a 190 µm wide by 125 µm high rectangular area centered around the midline of the tip of the heads and tails in size-matched control and *nr4A(RNAi)* animals (at least four animals per condition), through the entire thickness (DV axis) of the animal (*Figure 6—source data 2*). The numbers of EdU[+]; *collagen*[+] and EdU[+] cells were counted in size- and location-matched areas in control and *nr4A(RNAi)* animals through a 72 µm stack for head, posterior head, and tail regions, and a 33 µm stack in the trunk regions (*Figure 6—source data 3*, *Figure 6—figure supplement 1—source data 2*). Quantification of TUNEL[+] cells was done in size- and location-matched head tip areas in control and *nr4A(RNAi)* animals through the entire thickness (DV axis) of the animal (*Figure 6—figure supplement 1—source data 3*). Quantification of PCG expression domain shifts between control and *nr4A(RNAi)* animals was done in ImageJ (at least three animals per condition) (*Supplementary file 1D*). Before PCG domain quantification, animal eyes were occluded by black boxes to de-identify their RNAi condition when possible. Images were then randomized and the positions of the anterior or posterior boundaries of PCG expression domains were designated by another individual drawing lines at the boundaries. Distances between the midpoint of the boundary lines to head tip, anterior edge of original eyes, or tail tip were calculated and normalized to animal length. For *sFRP-1*, *wnt11-1*, and *wnt11-2*, curves following their expression boundaries were drawn and areas of their expression domains were calculated and normalized to the entire area of their respective animals. A one-tailed Welch's *t*-test was used to test for PCG boundary recession in *nr4A(RNAi)* animals. The presence of *wnt11-2* cluster of expression at the tail tip was blindly scored as present or absent, and a Fisher's Exact Test was used for statistical analysis. For anterior-pole-to-head-tip quantifications after one week of RNAi, the distance of each *notum*[+]; *collagen*[+] cell to the apex of the head was measured on Fiji/ImageJ. This distance was divided by the distance between the eyes within each animal for normalization (*Figure 8—source data 1*). Kruskal-Wallis test with Dunn's multiple comparisons test was used for

statistical analysis of anterior-pole-cell-to-head-apex distances between control and *nr4A(RNAi)* animals (at least five animals per condition). For newly incorporated eye cell quantifications, all *opsin*[+]; SMEDWI-1[+] cells in each eye were counted in control, 2-eye-pair *nr4A(RNAi)* animals, and 3-eye-pair *nr4A(RNAi)* animals (*Figure 9—source data 1*). Cells were included in the count if located within the eye itself or within approximately one cell diameter of the eye.

### RNA sequencing following *nr4A* RNAi

Heads and tails of animals subjected to 2, 3, 4, and 5 weeks of control or *nr4A* RNAi were surgically isolated and separately processed for RNA sequencing (*Figure 5A*), as described in Bulk RNA sequencing in Materials and methods. Three biological replicates of heads and tails were collected per time point, with six animals pooled within each biological replicate. Gene expression in *nr4A (RNAi)* heads and tails were compared with gene expression in control(RNAi) heads and tails, respectively, within each RNAi time point, as described in Bulk RNA sequencing in Materilas and methods. Sequencing data and DESeq analysis available in NCBI/GEO (accession GSE121048).

### Double RNAi

Three RNAi groups, control RNAi, control; *nr4A* RNAi, and *foxD; nr4A* RNAi were selected to test the effect of genetic pole ablation on the *nr4A(RNAi)* phenotype. Control RNAi used dsRNA against the *C. elegans unc-22* transcript. Animals in the *foxD; nr4A(RNAi)* group were subjected to only *foxD* RNAi for 6 weeks (two feedings/week) before the addition of *nr4A* RNAi. During this initial single RNAi period, animals in control and control; *nr4A(RNAi)* groups received only control RNAi, with *nr4A* RNAi added to the *control; nr4A(RNAi)* group at the same time as the *nr4A* RNAi in the *foxD; nr4A(RNAi)* group. At the end of the single RNAi period, the anterior poles from some animals were excised with a scalpel as described for anterior pole sequencing. These pole-cut animals were allowed to recover for 3 days before initiating the double RNAi. Double RNAi was performed for a total of 10 weeks (with two feedings/week) for intact animals and 12 weeks for pole-cut animals, with one-half of the total amount of dsRNA being from each kind dsRNA (see single RNAi in Materials and methods). Two biological replicates of double RNAi with uninjured animals and two biological replicates of double RNAi with pole-cut animals were performed.

### qPCR

Six trunk fragments per biological triplicate were isolated for each of the RNAi groups (control, control; *nr4A*, and *foxD; nr4A*). The uninjured and pole-cut animal groups were analyzed separately. Tissues were homogenized using Qiagen TissueLyser III, and RNA was extracted using TRIzol according to the manufacturer's protocol, and synthesized into cDNA with reverse transcriptase per manufacturer's protocol. qPCR primers for *nr4A* (*Supplementary file 1F*) were designed to amplify regions outside of the dsRNA sequences for the respective genes. Primers for the housekeeping gene *gapdh* (*Supplementary file 1F*) were used for expression normalization. Expression levels were calculated by the double delta-Ct method, with normalized *nr4A* expression levels in control; *nr4A*, and *foxD; nr4A(RNAi)* groups compared to normalized *nr4A* expression levels in the control(RNAi) group.

### EdU labeling

F-ara-EdU (Sigma) dissolved in DMSO at 50 mg/mL was mixed with liver paste (3:1 liver: planarian water) for a final concentration of 0.5 mg/mL. Animals were fed once with EdU and liver mix and fixed 8 days after as in in situ hybridization. Development was performed with click chemistry, as described (*Salic and Mitchison, 2008*; *Ji et al., 2017*).

### TUNEL assay

ApopTag Red In Situ Apoptosis Detection Kit (Millipore) was used for the TUNEL assay. Animals were fixed as for in situ hybridization, incubated overnight at 37°C in terminal deoxynucleotidyl transferase diluted in reaction buffer, washed in stop/wash buffer, rinsed in PBST (0.3% TritonX), and blocked for 30 min in PBSTx containing 5% heat-inactivated horse serum and 5% Western Block Reagent (Roche). Animals were then developed in anti-digoxigenin rhodamine conjugate diluted in block solution overnight at 4°C in the dark.

## Irradiation

Animals were given lethal irradiation dosage of 6000 rads using a Gammacell 40 dual $^{137}$cesium source.

## Data availability

Bulk and single-cell RNAi sequencing of planarian poles, and bulk RNA sequencing of *nr4A* RNAi animal heads and tails. Li, DJ, McMann, CL, Reddien PW (2018) available at the NCBI Gene Expression Omnibus (Accession no.: GSE121048).

## Acknowledgements

We thank Omri Wurtzel and Lauren Cote for help with RNA sequencing data analysis, Lauren Cote with NR4A phylogenetics and PCG domain quantification, and Lucila Scimone and Christopher Fincher for providing single-cell RNA sequencing data for analysis. We thank Kutay Deniz Atabay for guidance on eye progenitor quantification and Yong Uk Kim for help with graphics and illustrations. We are grateful to all members of the Reddien Lab for their input on the project and on the paper.

## Additional information

### Funding

| Funder | Grant reference number | Author |
| --- | --- | --- |
| Howard Hughes Medical Institute | Investigator | Peter W Reddien |
| National Institutes of Health | R01GM080639 | Peter W Reddien |
| National Institutes of Health | T32GM007753 | Dayan J Li |

The funders had no role in study design, data collection and interpretation, or the decision to submit the work for publication.

### Author contributions

Dayan J Li, Conceptualization, Data curation, Formal analysis, Investigation, Methodology, Writing—original draft, Writing—review and editing; Conor L McMann, Data curation, Formal analysis, Investigation, Writing—review and editing; Peter W Reddien, Conceptualization, Resources, Formal analysis, Supervision, Funding acquisition, Investigation, Writing—original draft, Project administration, Writing—review and editing

### Author ORCIDs

Dayan J Li (ID) https://orcid.org/0000-0001-8154-9019
Conor L McMann (ID) https://orcid.org/0000-0002-2264-4646
Peter W Reddien (ID) https://orcid.org/0000-0002-5569-333X

### Decision letter and Author response

Decision letter https://doi.org/10.7554/eLife.42015.040
Author response https://doi.org/10.7554/eLife.42015.041

## Additional files

### Supplementary files

• Supplementary file 1. Summary of data from bulk RNA sequencing, single-cell RNA sequencing, PCG domain quantifications, *nr4A* RNAi target screen, and phylogenetic analysis. (**A**) Anterior-pole-enriched genes in uninjured animals. (**B**) Anterior-pole-enriched genes in anterior blastema of 72hpa trunks. (**C**) Posterior-pole-enriched genes from single-cell sequencing. (**D**) Quantification of PCG and *mag-1* domain shifts in *nr4A* vs. control RNAi. (**E**) *nr4A*$^+$ muscle vs. *nr4A*$^-$ muscle SCDE. (**F**) Primers

used to clone *nr4A* and its target genes, and qPCR primers. (**G**) Nuclear receptor proteins and their accession numbers used in phylogenetic analysis.

DOI: https://doi.org/10.7554/eLife.42015.035

• Transparent reporting form

DOI: https://doi.org/10.7554/eLife.42015.036

### Data availability

Sequencing data and DESeq analysis have been deposited in GEO under accession code GSE121048.

The following dataset was generated:

| Author(s) | Year | Dataset title | Dataset URL | Database and Identifier |
|---|---|---|---|---|
| Li DJ | 2019 | Nuclear receptor NR4A is required for patterning at the ends of the planarian anterior-posterior axis | https://www.ncbi.nlm.nih.gov/geo/query/acc.cgi?acc=GSE121048 | NCBI Gene Expression Omnibus, GSE121048 |

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
