## [Decision Letter]

Thank you for submitting your article "Nuclear receptor NR4A patterns the ends of the planarian anterior-posterior axis" for consideration by *eLife*. Your article has been reviewed by two peer reviewers, including: Alejandro Sánchez Alvarado as the Reviewing Editor and Reviewer #1, and the evaluation has been overseen by Marianne Bronner as the Senior Editor.

The reviewers have discussed the reviews with one another and the Reviewing Editor has drafted this decision to help you prepare a revised submission.

Summary:

In this manuscript, Li, McMann, and Reddien use a combination of tissue fragments and single-cell RNA-sequencing to search for additional patterning regulators. Their experiments identify new genes expressed at the anterior and posterior poles, of which the authors proceed to analyze the function of a homolog of the orphan nuclear receptor NR4A. Through a sequencing approach, the authors identify factors enriched in anterior and posterior poles (including *nr4a*). They then describe a multi-eye phenotype after *nr4a(RNAi)*, as well as a wide range of polarity control gene perturbations along the A/P axis. Though several transcription factor genes have been shown to be important in polarity for planarians, the authors argue that the *nr4a(RNAi)* phenotype is distinct, suggesting a new class of patterning defects. However, iterative eye formation (toward the head) was also seen in longer-term *notum(RNAi)* animals (Hill and Petersen, 2018) and iterative eye formation (away from the head) was seen in nou-darake(RNAi) animals (Cebriá, et al). The authors also claim that the role of *nr4A* in patterning both ends of the AP axis is unique (subsection “*nr4A* as a new patterning gene”, first paragraph), yet *islet1* also affects both anterior and posterior poles (März et al. and Hayashi et al.). In that light, it is our collective opinion that it is not accurate to say that this is a novel phenotype or phenotype class. Although the data are well quantified for the most part, and the work is proficiently executed, key weaknesses as listed below need to be addressed fully before the paper can be considered further for publication in *eLife*.

Essential revisions:

The authors interpret their results to conclude a role of *nr4a* specific to the poles. Yet, the title of the article and the main claims that *nr4a* functions at the poles and affects A/P polarity are somewhat under-supported. An alternative interpretation that fits most (if not all) results is that *nr4a* acts globally, but that the special functions of poles cause the *nr4a(RNAi)* phenotype to be most obvious there first. It is our shared assessment that two general major weaknesses need to be addressed by the authors. The first class can be addressed by experimental approaches, while the second class can be addressed by rewriting parts of the manuscript to better reflect prior published work.

1) The authors claim that NR4A specifically affects A/P pole cells. Given the global muscle expression of *nr4a* (including in stem cells) and the broad expression of genes differentially expressed after *nr4a(RNAi)*, it seems that another logical model would be that *nr4a* is globally affecting muscle specification, progenitor migration, survival, or function. It could be that a global phenotype (e.g. premature differentiation of migrating progenitors would be consistent) first manifests itself in the A/P poles because these cells are so critical to body organization or because these cells have to migrate the longest distance.

a) To test whether *nr4a* is globally required for myogenesis, the authors should examine incorporation of new muscle cells in head, trunk, and tail with BrdU or another analog and either muscle progenitor or pan-muscle markers. These experiments would tell us if *nr4a* is important for muscle cell specification and incorporation into the muscle and – because it could show whether head, trunk, and tail are equally affected – would help to support the authors model that *nr4a* is functioning in a special way in the head/tail. Alternatively, if the authors' model supports asymmetric apoptosis or another route to explain the differences, this could be explored.

b) Figure 3A needs quantification. It could be informative to measure the% of muscle cells that express *nr4A* in the head vs. trunk vs. tail. This would help distinguish whether the apparent head enrichment is higher expression per cell or a higher percentage of *nr4a*+ cells or neither.

c) Is dorsal/ventral patterning affected by *nr4a(RNAi)*? The title suggests that only A/P polarity is affected, but *slit* appears to be abnormal (indicating medial/lateral perturbation and potentially other disorganization). Are *bmp4-1*^+^ cells in the correct place? The *nr4a* phenotype also looks reminiscent of the *bmp4-1(RNAi)* phenotype eye-wise.

d) Can the authors demonstrate head/tail enrichment of the genes that are downstream of NR4A? Three (of 100) are claimed to be head/tail enriched (subsection “*nr4A* regulates the expression of muscle-specific markers and head and tail PCGs”, second paragraph), but this isn't entirely convincing by eye. Is head/tail enrichment supported for this gene set in other available datasets (e.g. Stückemann, et al.)? If the ~100 genes differentially regulated after *nr4a(RNAi)* are largely not head/tail enriched, this could also support a model where *nr4a* is acting everywhere in some process that is first manifested phenotypically in the head/tail.

e) Did the authors test the function of other new pole genes in tissue patterning, such as the candidate secreted molecule *kallman1*? The section uncovering new pole genes makes a somewhat abrupt transition to *nr4a* given that this group has the leading expertise and screening all twelve anterior-pole-enriched genes requires minimal resources. I think a comment could be made about what work was performed that led to analysis or *nr4a*, especially if other anterior-pole genes did not produce phenotypes.

f) The authors should quantify the results in Figure 4C and Figure 6C (e.g. # of fibers in an orientation per unit distance as per Scimone, et al., 2017). This would boost the conclusion that muscles are structurally normal in *nr4A(RNAi)* animals at the early time point (6C) and that defects are head-specific later (4C).

g) Can the authors clarify the muscle phenotypes in Figure 4C? The text claims "reduced body wall muscle in the head" which seems clear from the 9w dorsal but not ventral images which-if anything-show stronger staining in the *nr4a(RNAi)* animal. Also the method for counting cells in this panel is unclear. Number of cells per what area/volume? What region was counted and how was the tip defined?

2) The presentation of data in Figure 5 and Figure 5—figure supplement 1 was difficult to follow and likely will be challenging for someone outside of the planarian field. A diagram or set of diagrams showing how each domain shifts and changes size (for the significant results only) would dramatically improve the reader's ability to integrate all of these different factors and get a global picture of what is happening after loss of *nr4A*.

a) Do trunk domains (e.g. *ndl-3, ptk-7*) change in size after *nr4A(RNAi)*? Since the animals were all cropped, it was hard to get a sense of this. Based on the other results, I suspect the trunk might be shortened, but it would be helpful to know for sure.

b) It would be helpful at the end of the section to recap and bring all of the data together into a coherent conclusion. Are all domains shifting toward the trunk? Is the trunk diminished due to expanding poles? Are the data incongruous with different trends for different molecules even in the same space? Are boundaries between domains more blended/overlapping after *nr4A(RNAi)* or is the overall setup of polarity genes the same, just distorted in ratios/sizes?

3) Could the authors expand their interpretation/description of where *nr4a* fits in the cell signaling and gene regulatory logic underlying anterior-posterior patterning in planarians? The last figure (Figure 7E) provides a nice description of the *nr4a* phenotype, but a mechanistic model (perhaps in addition to the existing summary) of how this transcription factor regulates patterning would be helpful. In the Discussion, the authors touch on known roles of NR4A homologs in ligand-independent processes. Given the striking difference between homeostasis and regeneration in *nr4a(RNAi)* phenotypes, do the authors think *nr4a* functions autonomously? Or, does *nr4a* function depend on regional signals? Is it possible to address this question experimentally? For example, what happens to *nr4a* expression after manipulating other genes involved in cell signaling, such as *ndk, fz5/8-4* or *wntA? nr4a* appears to function as a key intermediary and without its function, differentiation of pole cells shifts anatomically. It would be interesting to explore how the *nr4a* expression domain behaves in reciprocal experiments to those presented in Figure 5 (i.e., knocking down patterning genes and inspecting *nr4a* expression to determine if it changes).

4) *nr4a* (dd-Smed_v4 or 6_12229) appears to be differentially expressed following *myoD* RNAi (48 hpa). We did not peruse other gene lists, such as those derived from single muscle cell RNA-seq studies, but the *myoD* result indicates that the tissue fragment single-cell RNA-seq approach taken in this paper was important and necessary for uncovering additional pole genes. The authors could discuss whatever links may exist between this work and their previous studies.

---

## [Author Response]

Summary:In this manuscript, Li, McMann, and Reddien use a combination of tissue fragments and single-cell RNA-sequencing to search for additional patterning regulators. Their experiments identify new genes expressed at the anterior and posterior poles, of which the authors proceed to analyze the function of a homolog of the orphan nuclear receptor NR4A. Through a sequencing approach, the authors identify factors enriched in anterior and posterior poles (including nr4a). They then describe a multi-eye phenotype after nr4a(RNAi), as well as a wide range of polarity control gene perturbations along the A/P axis. Though several transcription factor genes have been shown to be important in polarity for planarians, the authors argue that the nr4a(RNAi) phenotype is distinct, suggesting a new class of patterning defects. However, iterative eye formation (toward the head) was also seen in longer-term notum(RNAi) animals (Hill and Petersen, 2018) and iterative eye formation (away from the head) was seen in nou-darake(RNAi) animals (Cebriá, et al). The authors also claim that the role of nr4A in patterning both ends of the AP axis is unique (subsection “nr4A as a new patterning gene”, first paragraph), yet islet1 also affects both anterior and posterior poles (März et al. and Hayashi et al.). In that light, it is our collective opinion that it is not accurate to say that this is a novel phenotype or phenotype class. Although the data are well quantified for the most part, and the work is proficiently executed, key weaknesses as listed below need to be addressed fully before the paper can be considered further for publication in eLife.

We agree with the reviewers that multiple sets of ectopic eyes can be found in *notum* and *ndk(RNAi)* animals, and that worded as such (emphasis on iteration), this is a similarity with the *nr4A(RNAi)* phenotype. However, the nature of the eye iteration phenotype is different in all three cases. This comment was helpful in underscoring the importance to us of comparing our phenotype to other known cases of pattern disruption in the manuscript – including with additional data (Figure 9A, B, Figure 9—figure supplement 1) and wording improvements (such as the in the Discussion) – to clarify the nature of the *nr4A(RNAi)* phenotype (and the fact that it is distinct from these other cases). In brief, in the case of *notum* RNAi, ectopic eyes appear anteriorly, whereas in the case of *nr4A* and *ndk* RNAi the ectopic eyes appear posteriorly. In the case of ectopic eyes in *ndk* RNAi the eyes appear in posterior regions (including far into the trunk and the tail) with ever increasing distance from the head tip and anterior pole over time (Cebriá et al., 2002, Figure 3—figure supplement 2A). By contrast, *nr4A* RNAi leads to the appearance of ectopic eyes restricted to the head region, at a confined distance from the head tip and the anterior pole (Figure 2A, Figure 3A, Figure 9, Figure 9—figure supplement 1). In the case of *nr4A* RNAi, the anterior pole moves posteriorly and the head tip to original eye distance shrinks (Figure 3A, Figure 7A, C, Figure 8A, B, Figure 9A, Figure 9—figure supplement 1), whereas this does not happen in *ndk* RNAi (Cebriá et al., 2002, Scimone et al., 2016). Our data therefore suggest that fundamentally different underlying cellular explanations exist for the *nr4A* and *ndk(RNAi)* phenotypes. With regards to genes impacting both ends of the animal, we agree that is interesting and important to discuss in the text that *islet1* can affect both poles (now mentioned in the Discussion). However, the *islet1(RNAi)* phenotype is also very different from the *nr4A(RNAi)* phenotype. The anterior and posterior patterning phenotype in *islet1* RNAi is reported as a regeneration phenotype that involves midline collapse (fused eyes in the head and fused ventral nerve cords in the tail), associated with disruption of *slit* expression domain in the head and complete abrogation of *wnt1* expression in the tail (Hayashi et al., 2011; März et al., 2013). *nr4A* RNAi results in a homeostatic, not regenerative phenotype that is different in anatomical pattern; for example it does not result in midline collapse or general loss of pole function.

We have now taken the *nr4A(RNAi)* phenotype to the extreme with continuous homeostatic *nr4A* inhibition for as long as 21 weeks (Figure 9A, Figure 9—figure supplement 1). We tracked the fate of the eyes of individual animals throughout the course of our RNAi and in several instances noted clear disappearances of the original eyes at the head tip (Figure 9A, Figure 9—figure supplement 1). We have also observed fusion events between older and newer eyes associated with crowding at the head tip (Figure 9A, Figure 9—figure supplement 1). This is consistent with our model of shifting PCG domains and decreased progenitor targeting to the head tip (Figure 9C), and is different from other patterning phenotypes previously reported.

Taken together, we feel that these features make *nr4A(RNAi)* phenotype a novel phenotype, and hope that new data and wording improvement clarify this phenotype. We understand that the word “class” may be interpreted as wholly distinct from all other patterning phenotypes and have removed it from our description to avoid that misconception.

Essential revisions:The authors interpret their results to conclude a role of nr4a specific to the poles.

We do not intend our wording to indicate a conclusion that a role of *nr4A* is *specific* to the poles. We do find, however, that the first cellular defect detected following *nr4A* RNAi is a shift in the anterior pole – perhaps this led to the misperception. Given this comment, we have revised the text carefully throughout to ensure this is not implied inadvertently in our wording.

Yet, the title of the article and the main claims that nr4a functions at the poles and affects A/P polarity are somewhat under-supported.

The title of our paper was, "Nuclear receptor NR4A patterns the ends of the planarian anterior-posterior axis," which was not intended to be interpreted as "functions at the poles" (we would reserve use of the word pole for the anterior/posterior pole cells themselves) or to mean "affects A/P polarity" (it does not affect A/P polarity so much as AP pattern). Instead, the intent is that *nr4A* function impacts the pattern of the head end and tail end, and *nr4A* function impacts the location of the anterior and posterior pole cells. To make this clearer we have re-worded the title to read: "Nuclear receptor NR4A is required for patterning at the ends of the planarian anterior-posterior axis."

An alternative interpretation that fits most (if not all) results is that nr4a acts globally, but that the special functions of poles cause the nr4a(RNAi) phenotype to be most obvious there first. It is our shared assessment that two general major weaknesses need to be addressed by the authors. The first class can be addressed by experimental approaches, while the second class can be addressed by rewriting parts of the manuscript to better reflect prior published work.1) The authors claim that NR4A specifically affects A/P pole cells. Given the global muscle expression of nr4a (including in stem cells) and the broad expression of genes differentially expressed after nr4a(RNAi), it seems that another logical model would be that nr4a is globally affecting muscle specification, progenitor migration, survival, or function. It could be that a global phenotype (e.g. premature differentiation of migrating progenitors would be consistent) first manifests itself in the A/P poles because these cells are so critical to body organization or because these cells have to migrate the longest distance.

We do not intend our wording to claim that NR4A specifically affects A/P pole cells (and have carefully edited the main text to avoid that misperception). Rather, *nr4A* is required to properly pattern the head and tail. We do claim, with support from data, that shifts in the pole cells are the earliest manifestations observed in the *nr4A(RNAi)* phenotype (Figure 7, Figure 7—figure supplement 1, Supplementary file 1D), which also includes broader disruptions in head and tail PCG expression domains as well as in differentiated tissues like the muscle and the epidermis (Figure 2, Figure 2—figure supplement 2, Figure 6, Figure 6—figure supplement 1, Figure 7—figure supplement 1, Supplementary file 1D). Additional data and data analyses to address specific points raised by the reviewers are below.

a) To test whether nr4a is globally required for myogenesis, the authors should examine incorporation of new muscle cells in head, trunk, and tail with BrdU or another analog and either muscle progenitor or pan-muscle markers. These experiments would tell us if nr4a is important for muscle cell specification and incorporation into the muscle and – because it could show whether head, trunk, and tail are equally affected – would help to support the authors model that nr4a is functioning in a special way in the head/tail. Alternatively, if the authors' model supports asymmetric apoptosis or another route to explain the differences, this could be explored.

To address this myogenesis question, we utilized EdU to analyze the production of new muscle cells in *nr4A* RNAi and incorporated the results and findings in the revised figures and main text of the manuscript (Figure 6D, E, Figure 6—figure supplement 1C, D).

We observed that at 9 weeks of RNAi the number of EdU; *collagen* double-positive cells (new muscle cells) was significantly lower in the head tip of *nr4A(RNAi)* animals compared to that in a similar region (by location and area) in the head tips of control animals (Figure 6D). In contrast, the number of new muscle cells was not significantly different in the posterior head region, or significantly reduced in the trunk or tail regions of *nr4A(RNAi)* animals compared to those in equivalent regions in control animals (Figure 6D). We also quantified the total EdU incorporation events (not just muscle) in the same head tip, posterior head, trunk, and tail regions (Figure 6—figure supplement 1C), and observed no significant differences between control and *nr4A(RNAi)* animals. If anything, both the number of new muscle cells and total EdU incorporation events in the posterior head region were slightly higher (though not statistically significant) in *nr4A(RNAi)* animals compared to control animals, suggesting a posterior accumulation of new cells and new muscle cells that do not make their way to the head tip.

Furthermore, we quantified the number of new muscle cells and total EdU incorporation events in the head tip area immediately anterior to the eyes in control animals and compared to those in the head tip area immediately anterior to the original (anterior-most) eyes and in the head tip area anterior to the newest (posterior-most) ectopic eyes in *nr4A(RNAi)* animals (Figure 6E, Figure 6—figure supplement 1D). Compared to the numbers in the control animal head tip areas, we observed that the number of new muscle cells and total EdU incorporation events were significantly lower in head tip areas anterior to the original eyes but not significantly different in head tip areas anterior to the newest eyes in *nr4A(RNAi)* animals. This is consistent with the head tip decay with respect to the original eyes observed in *nr4A* RNAi (Figure 2A, C, Figure 9, Figure 9—figure supplement 1).

We also assessed the degree of apoptosis at the head tips by quantifying the number of TUNEL+ cells in equivalent regions (by location and area) in control and *nr4A(RNAi)* animals (Figure 6—figure supplement 1E). We observed no significant increase in apoptosis in the head tips of *nr4A(RNAi)* animals that would contribute the head tip decay.

Together these data identify no general myogenesis defects in *nr4A* RNAi, and reveal a head-tip-specific effect in progenitor targeting. These data are discussed in the text and the Discussion.

b) Figure 3A needs quantification. It could be informative to measure the% of muscle cells that express nr4A in the head vs. trunk vs. tail. This would help distinguish whether the apparent head enrichment is higher expression per cell or a higher percentage of nr4a+ cells or neither.

We performed the suggested quantification of the proportion of muscle (*collagen*^+^) cells that express *nr4A* in the head tip, trunk region just anterior to the pharynx, and tail tip (Figure 4B). Whereas *nr4A* is expressed broadly, there is a significantly higher proportion of muscle cells expressing *nr4A* in the head tip and tail tip compared to that in the trunk.

c) Is dorsal/ventral patterning affected by nr4a(RNAi)? The title suggests that only A/P polarity is affected, but slit appears to be abnormal (indicating medial/lateral perturbation and potentially other disorganization). Are bmp4-1+ cells in the correct place? The nr4a phenotype also looks reminiscent of the bmp4-1(RNAi) phenotype eye-wise.

In regards to *bmp4, in situ* hybridization analysis of *bmp4* expression in new data did not reveal overt differences in its expression pattern between control and *nr4A(RNAi)* animals (Figure 3—figure supplement 3B). No disruption in the dorsal expression domain of *bmp4* was detected and no ectopic ventral *bmp4* expression was observed in *nr4A* RNAi. *bmp4* expression in the esophagus was stronger in *nr4A(RNAi)* animals compared to control animals, but there were no ectopic foci of *bmp4* expression. Although *bmp4* RNAi in some cases also results in ectopic eyes (Reddien et al., 2007; Molina et al., 2007; Adell et al., 2010), their locations with respect to the original eyes (medial, posterior) differ from the locations of the ectopic eyes (posterior and lateral) in *nr4A* RNAi. Furthermore, we observed iterative appearance of posterior ectopic eyes in *nr4A* RNAi that is not observed in *bmp4* RNAi (Figure 2A, Figure 9, Figure 9—figure supplement 1).

*slit* expression is more disorganized medial-laterally in the head and is reduced in the tail, but its midline expression domain was normal in the midbody (Figure 3A and Figure 3—figure supplement 3C). Given the roles for the poles in specifying the midline (Hayashi et al., 2011, März, et al., 2013, Scimone et al., 2014, and Oderberg et al., 2017), disruption of *slit* expression in the head and tail is consistent with the pole shifts and anterior pole disorganization observed in *nr4A* RNAi.

These data suggest that DV patterning as a whole is not affected by *nr4A* RNAi, and that the ectopic eyes in *nr4A* RNAi are likely not due to a DV patterning abnormality. The changes in the expression DV boundary markers (*laminB*, NB.22.1e) in the head and tail of *nr4A(RNAi)* animals (Figure 2C, D) reflect more internal shifts of anterior and posterior tissues that occur subsequent to similar shifts in head and tail PCG domains.

d) Can the authors demonstrate head/tail enrichment of the genes that are downstream of NR4A? Three (of 100) are claimed to be head/tail enriched (subsection “nr4A regulates the expression of muscle-specific markers and head and tail PCGs”, second paragraph), but this isn't entirely convincing by eye. Is head/tail enrichment supported for this gene set in other available datasets (e.g. Stückemann, et al)? If the ~100 genes differentially regulated after nr4a(RNAi) are largely not head/tail enriched, this could also support a model where nr4a is acting everywhere in some process that is first manifested phenotypically in the head/tail.

We improved the figure representation of images to more clearly show the expression of some targets at head/tail ends (Figure 5B, Figure 5—figure supplement 1A). We also analyzed the expression levels of the NR4A targets in different segments of the AP axis using expression data from Stückemann et al., 2017 and observed that one-third of them have enriched expression in the head and tail (Figure 5—figure supplement 1B). We note that *nr4A* RNAi also affected the expression of the broadly expressed epidermal gene dd_950 (*vim*) at the ends (Figure 5B, Video 1, 2). That some NR4A-regulated genes were expressed and affected more broadly than only at the ends does not necessarily contradict the model that *nr4A* is required for patterning processes occurring specifically at the ends of the AP axis. We find it interesting that some of the targets display expression at the ends of the axis and these are good targets for future work.

e) Did the authors test the function of other new pole genes in tissue patterning, such as the candidate secreted molecule kallman1? The section uncovering new pole genes makes a somewhat abrupt transition to nr4a given that this group has the leading expertise and screening all twelve anterior-pole-enriched genes requires minimal resources. I think a comment could be made about what work was performed that led to analysis or nr4a, especially if other anterior-pole genes did not produce phenotypes.

We did perform RNAi on the anterior pole genes shown in Figure 1B. We were very intrigued by dd_13188 (*kallman1*), but its RNAi did not yield a phenotype. Besides *nr4A* RNAi, dd_7911 (*ror1*) RNAi resulted in lateral ectopic eyes. We decided early on in this work to focus on *nr4A* because we found its patterning phenotype exciting.

f) The authors should quantify the results in Figure 4C and Figure 6C (e.g. # of fibers in an orientation per unit distance as per Scimone, et al., 2017). This would boost the conclusion that muscles are structurally normal in nr4A(RNAi) animals at the early time point (6C) and that defects are head-specific later (4C).

We quantified the number of circular and longitudinal fibers per unit distance, as in Scimone et al., 2017 (Figure 6B, Figure 6—figure supplement 1B). Whereas there was drastic and broad muscle fiber loss in the head tips of *nr4A(RNAi)* animals after prolonged RNAi (9 weeks), no significant reduction in the number of circular or longitudinal muscle fibers were detected in the trunk or tail of *nr4A(RNAi)* animals compared to control animals. At an earlier RNAi time point (4 weeks), no significant differences in the number of circular and longitudinal fibers per unit distance were detected in the head of *nr4A(RNAi)* and control animals (Figure 6—figure supplement 1B). This confirms our findings that the muscle fiber loss is a late manifestation of the phenotype, and that such loss is concentrated at the head tip, and not the midbody or the tail, which is consistent with the reduction in the number of *collagen*^+^ cells in the head (Figure 6C).

g) Can the authors clarify the muscle phenotypes in Figure 4C? The text claims "reduced body wall muscle in the head" which seems clear from the 9w dorsal but not ventral images which-if anything-show stronger staining in the nr4a(RNAi) animal. Also the method for counting cells in this panel is unclear. Number of cells per what area/volume? What region was counted and how was the tip defined?

Appropriate changes were made in the text to describe more specifically the loss of circular, longitudinal, and diagonal muscle fibers in the head (subsection “Long-term nr4A RNAi causes head-specific reduction in muscle progenitor incorporation”, first paragraph), and better quality images of ventral muscle fibers in 4- and 9-week control(RNAi) animals were included (Figure 6A). The method for quantifying *collagen*^+^ cells in Materials and methods has also been included in Figure 6C legend for reference (190mm x 125mm area through the entire DV axis) and the regions quantified are indicated on the figure (Figure 6C).

2) The presentation of data in Figure 5 and Figure 5—figure supplement 1 was difficult to follow and likely will be challenging for someone outside of the planarian field. A diagram or set of diagrams showing how each domain shifts and changes size (for the significant results only) would dramatically improve the reader's ability to integrate all of these different factors and get a global picture of what is happening after loss of nr4A.

We thank the reviewers for the suggestion. A figure has been created to integrate and summarize the significant PCG domain shifts that occur during *nr4A* inhibition (Figure 3D).

a) Do trunk domains (e.g. ndl-3, ptk-7) change in size after nr4A(RNAi)? Since the animals were all cropped, it was hard to get a sense of this. Based on the other results, I suspect the trunk might be shortened, but it would be helpful to know for sure.

We have included figures of body-wide *ndl-3* and *ptk-*7 expression and quantified the distance between the posterior boundary of the *ndl-3* expression domain and the tail tip as well as the AP length of the *ptk-7* expression domain, normalized to animal length (Figure 3—figure supplement 3A, Supplementary file 1D). We observed no significant differences in these measurements between *nr4A(RNAi)* and control animals (Figure 3—figure supplement 3A, Supplementary file 1D). Furthermore, we performed additional quantifications of the distance (normalized to animal length) of the posterior boundaries of *ndl-2, ndl-5*, and *wnt2* expression domains from the head tip and observed no differences between control and *nr4A(RNAi)* animals (Supplementary file 1D). This shows that while *nr4A* inhibition changes PCG expression domains at the ends of the AP axis, PCG expression in the trunk remains relatively intact.

b) It would be helpful at the end of the section to recap and bring all of the data together into a coherent conclusion. Are all domains shifting toward the trunk? Is the trunk diminished due to expanding poles? Are the data incongruous with different trends for different molecules even in the same space? Are boundaries between domains more blended/overlapping after nr4A(RNAi) or is the overall setup of polarity genes the same, just distorted in ratios/sizes?

We agree that this is a helpful analysis/summary to have in the paper and we have endeavored to accomplish this with a diagram as suggested (Figure 6D) and text additions to the Results and Discussion (subsection “Long-term nr4A RNAi causes head-specific reduction in muscle progenitor 328 incorporation", subsection “A phenotype that fails to reach a stable pattern”, fourth paragraph).

3) Could the authors expand their interpretation/description of where nr4a fits in the cell signaling and gene regulatory logic underlying anterior-posterior patterning in planarians? The last figure (Figure 7E) provides a nice description of the nr4a phenotype, but a mechanistic model (perhaps in addition to the existing summary) of how this transcription factor regulates patterning would be helpful. In the Discussion, the authors touch on known roles of NR4A homologs in ligand-independent processes. Given the striking difference between homeostasis and regeneration in nr4a(RNAi) phenotypes, do the authors think nr4a functions autonomously? Or, does nr4a function depend on regional signals? Is it possible to address this question experimentally? For example, what happens to nr4a expression after manipulating other genes involved in cell signaling, such as ndk, fz5/8-4 or wntA? nr4a appears to function as a key intermediary and without its function, differentiation of pole cells shifts anatomically. It would be interesting to explore how the nr4a expression domain behaves in reciprocal experiments to those presented in Figure 5 (i.e., knocking down patterning genes and inspecting nr4a expression to determine if it changes).

Our primary mechanistic model for how nr4A regulates head and tail patterning focuses on the cellular and patterning gene expression change steps that underlie the phenotype. We have now enhanced this cellular model by further study of eye progenitor targeting to progressively posterior eyes and anterior eye decay (Figure 9A, B, Figure 9—figure supplement 1), and by further characterization of PCG expression domains (Supplementary file 1D).

We investigated how nr4A might fit with other patterning genes by examining nr4A expression in animals undergoing *ndk, fz5/8-4, wntA*, and *foxD* RNAi and observed no overt changes in nr4A expression pattern when these PCGs were inhibited and when anterior patterning was disrupted (Figure 3—figure supplement 2). We also detected no significant differences in body-wide nr4A expression by qPCR in *foxD(RNAi)* animals with absent poles compared to control animals (Figure 3—figure supplement 2B). These data suggest that while nr4A regulates pole placement and head and tail PCG expression patterns, it is likely not itself regulated by regional PCG signaling and may function autonomously in muscle.

4) nr4a (dd-Smed_v4 or 6_12229) appears to be differentially expressed following myoD RNAi (48 hpa). We did not peruse other gene lists, such as those derived from single muscle cell RNA-seq studies, but the myoD result indicates that the tissue fragment single-cell RNA-seq approach taken in this paper was important and necessary for uncovering additional pole genes. The authors could discuss whatever links may exist between this work and their previous studies.

The decrease in *nr4A* expression (p-value = 0.021) following *myoD* RNAi at 48 hpa (Scimone et al., 2017) should be interpreted with caution. Such decrease is not observed at earlier *myoD* RNAi time points (0hpa, 6hpa, or 24hpa) (Scimone et al., 2017). The decrease in *nr4A* expression only at 48hpa in *myoD(RNAi)* animals could reflect a difference in the amount of regenerated tissue in control and *myoD(RNAi)* animals, since *myoD* RNAi results in the complete block of regeneration after amputation, including the absence of both poles.

Our head tip fragment RNA sequencing approach offered a more precise spatial isolation of regions/cells to be sequenced than the single-cell RNA sequencing strategies in Scimone et al., 2016, and Fincher et al., 2018, which compared gene expression levels in broader anterior versus posterior segments. Furthermore, our sequencing of regenerating blastemas also aided in the identification of regionally enriched genes when paired with the uninjured animal data. In addition, due to the limited depth of single-cell RNA sequencing, we believe our bulk, regionally specific RNA sequencing strategy was better able to identify genes with low expression levels in uninjured animals, like *nr4A*. In fact, only ~10% of the single-muscle cells in Fincher et al., 2018, had detectable levels of *nr4A* expression, and most of the read counts ranged from 1-4. We have included new SCDE analysis of the *nr4A*^+^ muscle vs. *nr4A*^-^ muscle from Fincher et al., 2018, which showed that *collagen*-encoding genes were enriched in *nr4A*^+^ muscle, but no muscle subtype transcription factor genes, like *myoD* or *nkx1-1*, were differentially expressed (Supplementary file 1E).

We have also linked our findings with the muscle cell subtype clustering in Scimone et al., 2018, and showed that *nr4A* is expressed broadly in body-wall muscle, but not in intestinal or pharyngeal muscle (Figure 4E). Additionally, we showed that *nr4A* expression within body-wall muscle cluster is not concentrated preferentially in circular, longitudinal, or DV fibers (Figure 4—figure supplement 1), which is consistent with our finding that *nr4A* inhibition leads to general fiber loss at the head tip.